# Noisy Ostracods: A Fine-Grained, Imbalanced Real-World Dataset for Benchmarking Robust Machine Learning and Label Correction Methods

**Jiamian Hu**[1], **Yuanyuan Hong**[1], **Yihua Chen**[1,2], **He Wang**[1,3], **Moriaki Yasuhara**[1]

[1]The University of Hong Kong
[2]The University of Tokyo
[3]Nanjing Institute of Geology and Palaeontology, CAS
{jiamianh, u3001143, yihuaac}@connect.hku.hk, hwang@nigpas.ac.cn, yasuhara@hku.hk

## Abstract

We present the Noisy Ostracods, a noisy dataset for genus and species classification of crustacean ostracods with specialists' annotations. Over the 71466 specimens collected, 5.58% of them are estimated to be noisy (possibly problematic) at genus level. The dataset is created to addressing a real-world challenge: creating a clean fine-grained taxonomy dataset. The Noisy Ostracods dataset has diverse noises from multiple sources. Firstly, the noise is open-set, including new classes discovered during curation that were not part of the original annotation. The dataset has pseudo-classes, where annotators misclassified samples that should belong to an existing class into a new pseudo-class. The Noisy Ostracods dataset is highly imbalanced with a imbalance factor $\rho = 22429$. This presents a unique challenge for robust machine learning methods, as existing approaches have not been extensively evaluated on fine-grained classification tasks with such diverse real-world noise. Initial experiments using current robust learning techniques have not yielded significant performance improvements on the Noisy Ostracods dataset compared to cross-entropy training on the raw, noisy data. On the other hand, noise detection methods have underperformed in error hit rate compared to naive cross-validation ensembling for identifying problematic labels. These findings suggest that the fine-grained, imbalanced nature, and complex noise characteristics of the dataset present considerable challenges for existing noise-robust algorithms. By openly releasing the Noisy Ostracods dataset, our goal is to encourage further research into the development of noise-resilient machine learning methods capable of effectively handling diverse, real-world noise in fine-grained classification tasks. The dataset, along with its evaluation protocols, can be accessed at `https://github.com/H-Jamieu/Noisy_ostracods`.

## 1 Introduction

Ostracods are micro-crustaceans inhabiting in marine, non-marine and some semi-terrestrial habitats[1]. Their calcified shells preserved in the sediments provided rich material for paleoenvironmental reconstruction, ecological monitoring and bio-diversity analyses [2]. However, conducting such studies requires massive efforts in counting, sorting, and identifying ostracods [3]. With advancements in image acquisition technologies and deep learning methods [4, 5, 6], building an automatic identification system for ostracods has become a feasible solution to reduce human labor. Such a system necessitates a large amount of annotated data. In our practical work, the annotated dataset also serves as a taxonomy reference material for teaching. In these applications, any incorrect taxonomy

38th Conference on Neural Information Processing Systems (NeurIPS 2024) Track on Datasets and Benchmarks.

(noise) discovered in the dataset raises user concerns about the trustworthiness of the identification results from the models trained on this data. Therefore, the need for robust, trustworthy machine learning algorithms and cleaned datasets is critical for the adaption of deep learning methods in the taxonomy field.

Creating a clean dataset is both labor-intensive and time-consuming. Before the current version of Noisy Ostracods, we manually checked and retook corrupted photos over several rounds in a two-year period. Despite these efforts, new errors continue to be discovered by users, motivating us to conduct an extensive study on Learning with Noisy Labels (LNL) methods in the hopes of finding effective solutions. LNL is an active research field [7, 8] that aims to train robust machine learning models in the presence of label noise. In addition to making the training process more resilient to errors [9, 10, 11], popular approaches include correcting label errors using learned features [12] and model activations [13]. In this study, we present the Noisy Ostracods dataset, which contains 71,466 annotated instances distributed across over 100 species and 70 genera. We manually corrected 20% of the data at the genus level to provide a reference for assessing the performance of LNL and label correction methods. We focus on two research questions:

- **Robustness Against Label Noise**: How robust are LNL methods compared to standard Cross Entropy training on noisy data in the Noisy Ostracods dataset?
- **Label Correction Effectiveness**: What proportion of errors in the dataset can be detected by label correction methods compared to a cross-validation-based baseline (Algorithm 1)?

Despite our efforts to utilize these advanced LNL methods, they fell short of our expectations in fully addressing the challenges posed by our dataset. This motivated us to publish the Noisy Ostracods dataset and our experimental findings. Our aim is to provide a meticulously annotated, fine-grained benchmark dataset to facilitate further research in this area and encourage the development of more robust solutions for handling noise in fine-grained image classification tasks. Such methods would be beneficial for creators of real-world fine-grained taxonomy datasets, increasing efficiency and reducing the labor required for dataset creation.

## 2 Related works

**Fine-grained Datasets for Taxonomy and Their Quality**    Some of the most well-known image datasets for taxonomy are CUB-200-2011 [14] and Flower102 [15]. While Flower102 does not explicitly state the process used to ensure label quality, the CUB-200-2011 dataset involved domain experts to verify and ensure label accuracy. In recent years, there has been a significant increase in community-labeled large-scale datasets, such as iNaturalist [16], TREEOFLIFE-10M [17], and Pla@ntNet [18]. However, the proportion of expert-annotated data in these large-scale datasets is relatively small compared to the vast amounts of web-sourced data and facing large elimination ratio. Endlessforams [19] provides 34,640 expert-annotated images of foraminiferans, ensuring quality by including only those samples that achieved an agreement ratio of over 75% among three experts, resulting in the elimination of over 100,000 samples. Similarly, BIOSCAN-1M [17] includes 25% of the data labeled at the genus level and 8% at the species level out of its total 1 million images. The BIOSCAN dataset faces similar challenges to ours during annotation, such as noise from low-quality images and significant class imbalance. These issues highlight that the challenges encountered by the Noisy Ostracods dataset, such as label noise and imbalanced classes, are common across taxonomy datasets used by biologists. Addressing these challenges is crucial for improving the quality of fine-grained classification in biological research. Additionally, reducing the waste of images through better annotation practices and noise reduction techniques can significantly enhance dataset efficiency, ensuring that more images are usable and valuable for training robust models.

**Real-world Noisy Datasets**    Real-world noisy datasets are relatively rare in the literature. CIFAR-10N [20] and CIFAR-10H[21] are human re-annotated versions of CIFAR-10 [22] designed to model real-world noise. Clothing1M [23] is a web-scale dataset crawled from search engines, featuring noisy labels. Mini-ImageNet [24] is another human re-annotated noisy dataset, created from a subset of ImageNet. Compared to these common benchmarks, the Noisy Ostracods dataset features a higher proportion of class imbalance and more fine-grained classes. Furthermore, the target users of the cleaned ostracods dataset are domain experts, which imposes additional importance on error cleaning to ensure the dataset's reliability and usefulness.

# 3 The Noisy Ostracods dataset

The Noisy Ostracods dataset comprises ostracods collected from Hong Kong sediment samples over the last 10 years. These samples were used to assess the anthropogenic impact on Hong Kong marine environments [3, 2]. The taxonomy of the ostracods were identified by expert researchers during studies conducted over the past decade. We photographed the samples using VHX-7000 microscope [25] with 40x-300x magnification. We iteratively annotated, trained YOLO[4], refined annotations, removed errors, and retrained stronger YOLO models to crop ostracods from backgrounds. The annotation process primarily involves mapping taxonomy records by the expert to the corresponding photographs using fixed programs. Ultimately, the annotated dataset includes 71,466 specimens of ostracods across 78 *annotated* genera and 138 *annotated* species.

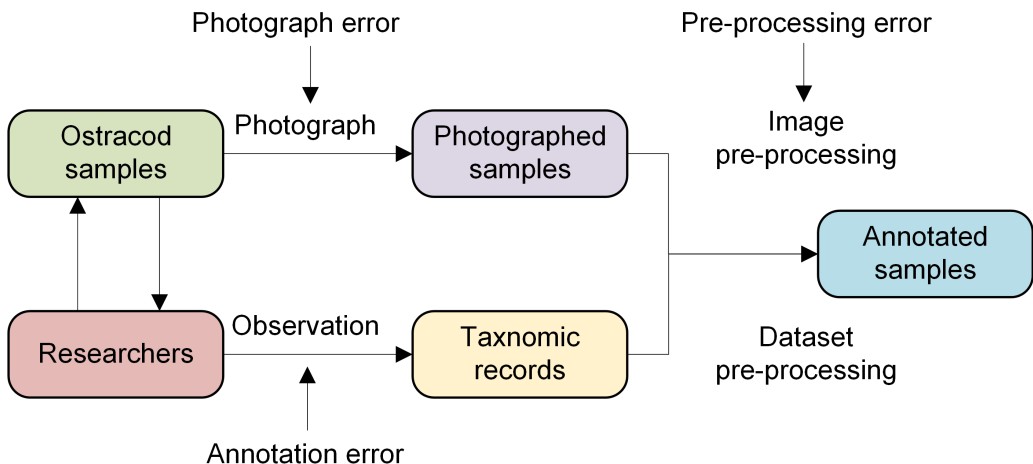

Figure 1: A simplified data collection and annotation process for the Noisy Ostracods dataset.

## 3.1 Noise source analyses

From the process described in 1, the noises in the Noisy Ostracods dataset is a aggrated result from original annotator, photograph process and preprocessing of the data. Different from the majority of the image datasets [26, 22, 20, 23, 24] annotating using the digital photos, the annotations of the dataset is created from the observation of the object being photographed directly. As a result, any photographic failure leads to noisy data. For example, photos (a) and (e) in Figure 2 show how excessive brightness—either too low or too high—can obscure key features, creating noise. Besides the photograph process, the preprocessing steps can also introduce errors. Even though the YOLO detectors are trained and refined through multiple iterations to ensure detection quality, some errors are unavoidable. The photo i in Figure 2 shows an example of such an error. These errors are rare in the dataset, with fewer than ten occurrences identified so far.[1] Lastly, the dataset contains annotation errors, including label flipping between classes. Sources of these errors may include misidentification of similar species, typographical errors in the records, and mismatching specimens to their corresponding recording files, among others. These diverse sources of noise highlight the complexity and challenges inherent in creating a high-quality, annotated dataset for fine-grained classification tasks.

## 3.2 Noise type in detail

We grouped noise types into two categories: feature errors and label errors. Empirically, to create a cleaned dataset, feature errors often result in deletion, whereas label errors lead to re-labeling. Label

---

[1]For researchers interested in the impact of detection preprocessing errors, we have also provide the Noisy Ostracods 2022 dataset in the repository. This dataset was created using earlier iterations of the YOLO detectors.

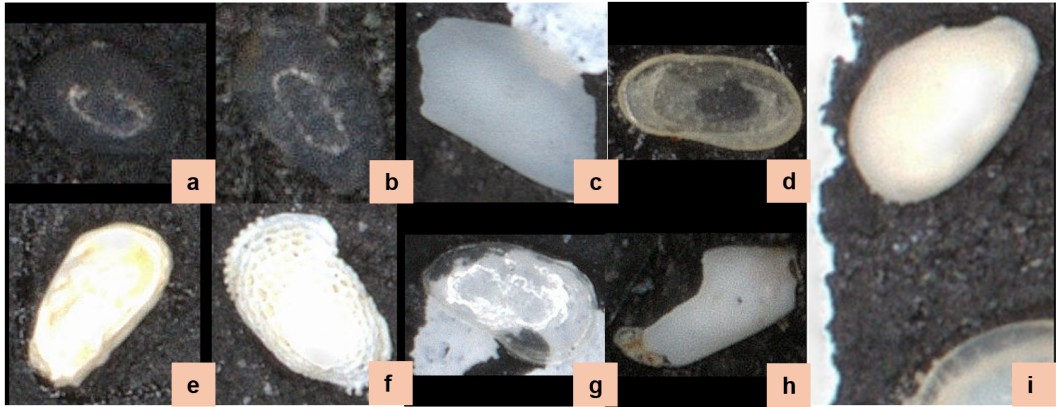

Figure 2: Selected noisy images from the Noisy Ostracods Dataset. a. Photograph error; b. Photograph error; c. Fragmentation error; d. Position error; e. Photograph error; f. Photograph error; g. Photograph error; h. Fragmentation error; i. Preprocess error.

error has been well studied in the literature [10, 11, 13, 20, 12, 27, 8, 7] including assumptions such as class-dependent, instance-dependent and annotator specific types. In contrast, feature errors are less frequently mentioned in LNL literature. These errors represent the loss or corruption of image features, making it difficult to assign a suitable class [8, 28].

**Feature error**     Define the noisy dataset for the taxonomic task as $\mathcal{D} = \{(\tilde{x}_i, \tilde{y}_i)\}_{i=1}^n$ where $n$ is the number of data points, $\tilde{x}_i$ and $\tilde{y}_i$ is the $i$-th image and its label. The given label $\tilde{y}_i \in \tilde{m}$ in $\tilde{m}$ annotated classes. Assume $m^* := \{m^* \mid \mathcal{D}\}$ is the set of all possible ground truth classes of taxonomy given the dataset. In Noisy Ostracods, $m^*$ includes all possible *true* genera and species of ostracods in the dataset. The negative class is represented by $\sim m^*$, the complementary class of $m^*$. Given $\mathcal{P}^*$ as the optimal model mapping features to the label space where $\mathcal{P}^*(y_i = m_i \mid x) = 1$ when the ground truth label of $x_i$ is $m_i$, the feature error can be defined as an out-of-distribution error given corrupted data $\tilde{x}_i$ such that:

$$\sup[\mathcal{P}^*(y_i \in m^* \mid \tilde{x}_i)] < \mathcal{P}^*(y_i \in \sim m^* \mid \tilde{x}_i). \tag{1}$$

Under such circumstances, it is risky to assign any label within the ground truth set $m^*$ for corrupted image $\tilde{x}_i$. To make the identification result reliable, the best label for such a sample could be *negative* ($\sim m^*$) rather than any genus/species of ostracods. The discovered types of feature error in the Noisy Ostracods dataset are:

- **Photographic error:** The photo taken for the ostracods does not reflect the key distinguishable features of the genus/species the specimen was labeled as. Examples are photos a, b, e, f, g in Figure 2.

- **Fragmentation error:** The shells of the ostracods are broken and missing the key distinguishable part of the labeled genus/species. Figure 2 c and h illustrate this case.

- **Position error:** The shell of the ostracods is not organized in an ideal manner for identification (e.g., up-side-down). Such a position could hide the key distinguishable features for taxonomy, making the identification vague. Photo d of Figure 2 shows this case.

- **Preprocess error:** As shown in photo i of Figure 2, this type of fault is introduced by detectors gave false areas of interest during preprocessing.

Different error types require different remediation approaches. Photographic and preprocessing errors can be corrected through re-imaging and re-annotation, respectively. However, specimens with fragmentation or position errors must be removed from the dataset. For efficiency, we implemented all corrections after completing the error detection process. Consequently, all feature errors were removed in the current cleaned version of the dataset.

**Label error**    Label error occurs when the given label $\tilde{y}_i$ is not the ground truth label $y_i$ and $y_i \in m^*$. Note that $\tilde{y}_i$ may belong to a pseudo classes in $\sim m^*$. Following are the discovered label errors in the Noisy Ostracods dataset:

- **Pseudo classes:** The dataset contains multiple pseudo classes as a result of typos and different usage standards of *Confer* (cf.) and *Affinis* (aff.) to mark the comparability between species. For instance, in the dataset, the genus *Cyprideis* is a typo of *Cytherois*. Additionally, *Callistocythere aff. reticulata* and *Callistocythere cf. reticulata* exist in parallel, pointing to nearly identical specimens. This type of error is highly project-specific, more details discussed in the Appendix G.

- **Mixed classes:** Due to the lack of usage as environmental indicators [3], the *Paradoxostom-aid* class is a mixture of the genera *Paradoxostoma* (Pd.), *Paracytherois* (Pc.), *Neocythero-morpha* (N), and *Xiphichilus* (X.). The ratio for the images with noise labels among the four genus is: $Pd. : Pc. : N. : X. = 1257 : 98 : 24 : 87$. After cleaning, around 75% of the originally labeled *Paradoxostomatid* images were assigned to the other three genera. Causing the presence of the majority wrong issue in the class. The details on can be found in the AppendixD.

- **New classes:** We discovered one new unnamed genus, during manual cleaning of the dataset. This suggests that the noise in the dataset is open-set. However, we removed those specimen belong to new genera to reduce the complexity of the project.

- **Hard classes:** Some species are hard to distinguish from each other from pictures alone. One example in our dataset is *Pistocythereis bradyi* and *Pistocythereis subovata*, which are nearly indistinguishable from the pictures alone. One possible solution is allowing multi-label classification results.

The diverse types of label noise indicate that label curation for the Noisy Ostracods dataset is challenging. The default solution for above error type is to re-label the images to its corresponding true classes. Under a single-label multi-class classification task, given the presence of *pseudo class* errors, *mixed class* errors, and *new classes*, the number of ground truth classes in the dataset is:

$$|m^*| := |\{m^* \mid \mathcal{D}\}| = |\tilde{m}| - |m_{\text{pseudo}}| + |m_{\text{new}}| \tag{2}$$

where $|m_{\text{pseudo}}|$ is the number of pseudo classes and $|m_{\text{new}}|$ is the number of newly discovered classes. From equation (2) we could infer that the final classes need to consider both out-of-distribution classes and pseudo classes. Therefore we emphasis the genus and species provided by the dataset is *annotated* rather than stating they are the ground truth genus/species.

### 3.3    Data split and cleaning

**Data split**    The data was initially split into train, test, and validation sets before cleaning. Our goal was to split the data into 80% training, 10% validation, and 10% test sets, while preserving the genus/species distribution of the original dataset. For each class, 10% of the data were sampled to validation set and test set to achieve this balance. We moved all classes with fewer than 10 samples into the training set, hoping to train the model on more diverse data. This resulted in a greater number of genus and species being represented in the training set compared to the test and validation sets. Specifically, the training set includes 78 genera and 138 species, whereas the test and validation sets include 51 genera and 81 species. Additionally, a negative class includes randomly cropped backgrounds from the micro-fossil slides and Foraminiferas from the Endless Forams[19] is added to the dataset.

**Cleaning**    To manage cleaning time, we first checked the test and validation sets at the genus level. This involved manually examining the dataset and discussing ambiguities with at least two experts. Unresolved ambiguities led to data deletion (Figure 5). Out of 14,320 samples, 795 noisy samples were identified, resulting in a confirmed noise ratio of 5.58%. Of these, 475 samples with feature errors were deleted, and the remaining 320 were re-labeled to the correct genus. Transition matrices are in Appendix 9.

# 4   Experiments

We conducted experiments on two popular directions of Learning with Noisy Labels (LNL): robust training on noisy data and label error detection. Robust training aims to enhance the model's tolerance to errors in the training set by leveraging loss regularization, optimization processes, and sample selection during training [8, 7]. In contrast, label error correction methods focus on detecting and correcting label errors in the training set [13, 29], with some methods providing suggestions for the correct labels [12, 30]. We adapted the official implementations of all selected methods for the following experiments with minimal changes.

## 4.1   Robust Learning Methods

To ensure fair comparisons, we used two fixed backbones, ResNet-50 [5] and ViT-B-16 [6], both pre-trained on ImageNet [31]. We evaluated the effectiveness of the small loss trick [9] on our dataset by implementing Loss-Clip, Co-teaching [10], and Co-teaching Plus [11]. Loss-Clip filters out a proportion of losses larger than $\tau = 0.05$. We also included MixUp [32] and CutMix [33] for their implicit regularization benefits [7], applying them randomly with equal probability. Additionally, we selected DivideMix [34] for its strong performance on CIFAR-10N [20], and Sharpness-Aware Minimization (SAM) [35] for its reported robustness to label noise [36]. The Confident Learning (CL) model involves training Cross-Entropy (CE) loss models on cleaned samples identified by Confident Learning, with more details discussed in the next section. Part-level Labels (PLM) [37] leverages image crops to mitigate false labels. We included Subclass-Dominant Label Noise (SDN) [38] as it addresses a specific noise pattern present in our dataset: ostracods have up to 9 juvenile stages that differ from adult forms, creating natural subclasses. For instance, juvenile stages of *Alocopocythere* and *Neocytheretta* are often confused, while their adult forms are more distinct. Label Wave[39] was added for its mechanism of finding better early stopping points to avoid memorization of noisy instances with out providing validation set. Notably, it presents an interesting contrast with SDN: while SDN recommends late stopping for subclass-dominant errors, Label Wave advocates for early stopping. We evaluated the effectiveness of noise transition matrix-based methods [27, 40] by implementing the widely-adopted noisy posterior approach [41, 42, 43]. In label noise learning (LNL), the noise posterior is typically formulated as:

$$P(\tilde{y}_i|x_i) = T(x)^T P(y_i|x_i) \tag{3}$$

where $T(x)$ represents the noise transition matrix. Rather than estimating the transition matrix, we utilized the actual noise transition matrix derived from our manual validation process (Figure 9). This approach allowed us to directly assess the effectiveness of class-dependent noisy posterior by adjusting the model's predicted probabilities using the true transition matrix. However, this method was constrained to training only on the 51 genera present in both the transition matrix and validation set, as described in the data split section. Consequently, the results from this method are not directly comparable with other approaches that utilize the full dataset. All models were trained for 300 epochs, repeated five times using the same hyperparameters, with the best run reported. Hyperparameters and training details are available in Appendix F.

**Results**   Overall, as shown in Table 1, the naive cross-entropy (CE) training outperformed other methods in the majority of metrics in both backbones. When comparing backbones, ResNet-50 generally outperformed ViT-B-16 across most metrics within the same methods.

For accuracy, only PLM achieved a slightly higher accuracy of 96.77% and 96.36% accuracy compared to the naive CE training with a ResNet-50 backbone. However the difference between the accuracy of CE and PLM just around 1%. Expect the transition matrix whose result is not directly incomparable, the difference in accuracy between the best and worst performers with the ResNet-50 backbone is just 2%, suggesting that all the models offered a similar level of performance. With the ViT-B-16 backbone, PLM and CE training clearly dominates, and this observation is consistent across metrics.

In terms of Precision, Recall, and F1-score, the naive CE training method consistently outperformed all other methods. A key finding from the results is that the selected robust methods did not provide additional robustness against noisy data on the Noisy Ostracods dataset when pre-trained weights on ImageNet were used. This suggests that for the Noisy Ostracods dataset or datasets with similar characteristics, fine-tuning on ImageNet pretraining weights can provide good robustness. This

Table 1: Experiments on different models. The maximum accuracy round of the methods over 5 runs are reported. Acc: Accuracy; P@: Precision; R@: Recall; F1@: F1 Score. The table with mean and variances are in AppendixH

| Method | ResNet-50 | | | | ViT-B-16 | | | |
|---|---|---|---|---|---|---|---|---|
| | Acc | P@ | R@ | F1@ | Acc | P@ | R@ | F1@ |
| CE | 95.98 | **88.50** | **77.80** | **79.51** | 95.03 | **83.31** | **75.30** | **76.64** |
| CE+mixup | 95.11 | 74.96 | 69.83 | 69.42 | 90.33 | 57.99 | 51.95 | 52.98 |
| CL | 95.16 | 80.97 | 69.01 | 71.51 | 90.62 | 57.48 | 53.27 | 53.75 |
| Loss-clip | 95.79 | 81.69 | 72.78 | 74.09 | 93.71 | 72.14 | 64.27 | 65.26 |
| Co-teaching | 95.79 | 79.01 | 71.79 | 73.23 | 94.71 | 77.77 | 71.69 | 72.37 |
| Co-teaching+ | 96.19 | 84.09 | 77.57 | 78.21 | 91.82 | 66.04 | 63.45 | 63.65 |
| DivideMix | 95.50 | 53.42 | 57.33 | 54.86 | 84.79 | 23.00 | 28.22 | 24.92 |
| SAM | 94.84 | 78.93 | 68.08 | 70.60 | 93.51 | 77.89 | 64.11 | 69.82 |
| PLM | **96.77** | 77.77 | 72.74 | 73.46 | **96.36** | 81.28 | 75.17 | 76.41 |
| SDN | 95.11 | 78.13 | 71.31 | 72.04 | 92.84 | 76.49 | 62.16 | 64.96 |
| LW | 95.50 | 86.59 | 74.28 | 76.93 | 94.65 | 77.15 | 69.61 | 71.35 |
| Transition* | 84.22 | 61.80 | 55.33 | 56.49 | 83.28 | 62.64 | 52.95 | 54.18 |

Transition matrix is trained on the dataset containing only 51 classes that have more than 10 photos. The result is not directly comparable with other results.

observation aligns with the literature, which suggests that using pre-trained weights can make fine-tuned models more robust [44].

Despite being trained on a reduced subset of 51 classes (each with more than 10 samples), the transition matrix approach significantly underperformed, achieving only 84.22% and 83.28% accuracy with ResNet-50 and ViT-B-16, respectively. This unexpectedly poor performance suggests that the assumption of class-dependent noise transitions may not hold for the Noisy Ostracods dataset, indicating more complex noise patterns than traditional class-dependent noise models can capture.

Our experiments reveal two significant findings about learning with noisy labels in real-world scenarios. First, the standard CE training achieved an error rate of 4.02%, which is notably lower than the dataset's noise rate of 5.58%. This suggests that modern deep learning architectures, when pre-trained on large-scale datasets like ImageNet, can effectively learn from noisy labels without specialized techniques. Second, and perhaps more crucially, none of the evaluated LNL methods provided substantial improvements over basic CE training, despite their reported success on synthetic noises in CIFAR-10. This performance gap between synthetic and real-world scenarios highlights the importance of including more real-world noise benchmarks such as the noisy ostracods dataset could help to the development of generalizable robust machine learning methods.

## 4.2 Label correction methods

While the previous section focused on model performance with noisy labels, a more critical aspect for taxonomic research is the ability to identify and correct labeling errors in existing datasets. After meticulously cleaning our test and validation sets, we conducted a detailed study comparing the efficacy of various label error correction methods. By comparing the label correction proposed by the model with the manual verified correction, the performance of the methods can be accessed. Label correction is closer to the actual application of LNL methods in our taxonomy research pipeline to acquire clean taxonomy dataset for taxonomy reference. We began by establishing a straightforward baseline: Naive-Ensemble-Cross-Validation (NECV) (Algorithm 1). The idea of NECV is not novel, similar ideas were presented in the literature for multiple times[45, 46, 40]. In NECV, we divided the dataset into $S = 10$ equal-sized splits. For each split, we trained $T = 14$ different models on the dataset, excluding the split being held out. The models used in this study are: ResNet-50, ResNet-152, ViT-B-16, ViT-L-16, ConvNext-large [47], RegNet-y-16bf [48], EfficientNet-v2-s [49], EfficientNet-V2-L, MnasNet [50], MaxViT [51], Swin-V2-B [52], MobileNet-V3 [53], ResNeXt-50 [54], and ShuffleNet [55]. This ensemble approach results in 14 predictions (votes) for each sample in the held-out split. We then calculated the agreement ratio, which is the proportion of the 14 votes that match the original label $\tilde{y}$ for each sample. If the agreement ratio is less than $\tau = 0.5$, the sample is

marked as noisy. This threshold reflects the assumption that if the majority of the ensemble disagrees with the original label, it is likely erroneous.

---

**Algorithm 1** Naive-Ensemble-Cross-Validation

---

**Require:** Dataset $\mathcal{D} = \{(\tilde{x}_i, \tilde{y}_i)\}_{i=1}^n$, number of splits $S$, set of models $T$, threshold $\tau$

    **return** Noisy samples $\chi$

    Split the dataset $\mathcal{D}$ into $S$ splits: $\{\mathcal{D}_i\}_{i=1}^S$

    Initialize $\chi = \emptyset$

    **for** $i = 1$ to $S$ **do**

        Treat $\mathcal{D}_i$ as the validation set

        Train each model $t \in T$ on the dataset excluding $\mathcal{D}_i$: $\mathcal{D} \setminus \mathcal{D}_i$

        **for** each data point $(\tilde{x}, \tilde{y})$ in $\mathcal{D}_i$ **do**

            $c \leftarrow 0$

            **for** each model $t \in T$ **do**

                Obtain prediction $\hat{y}$ from model $t$ on data point $\tilde{x}$

                **if** $\hat{y} = \tilde{y}$ **then**

                    $c \leftarrow c + 1$

                **end if**

            **end for**

            Calculate the agreement ratio for $(\tilde{x}, \tilde{y})$: agreement_ratio$(\tilde{x}, \tilde{y}) = \frac{c}{|T|}$

            **if** agreement_ratio$(\tilde{x}, \tilde{y}) < \tau$ **then**

                $\chi \leftarrow \chi \cup \{(\tilde{x}, \tilde{y})\}$

            **end if**

        **end for**

    **end for**

    **return** $\chi$

---

Confident Learning (CL) is a widely used baseline for classification label correction [13, 56, 57, 12]. CL utilizes the activation layer outputs from models to identify noisy samples. For our study, we provided CL with the activation layers of the 10 ResNet-50 models trained during NECV, with each model contributing activations for its corresponding excluded split.

SimiFeat [12], a top-performing method from the recent AQuA[56] benchmark, was also selected for comparison. SimiFeat leverages feature representations from pre-trained models to detect label noise. We employ two methods from SimiFeat: majority voting, which flags samples as noisy if the majority of their nearest neighbors disagree with their labels, and ranking, which scores each sample based on HOC[58] and identifies the lowest-scoring samples as noisy. Following the authors of SimiFeat, we used CLIP [59] as the feature extractor, selecting the strongest CLIP-ViT-L-14@336 backbone for our study. Additionally, we explored the potential of self-supervised feature extractors for SimiFeat. We selected Masked Auto-Encoders [60] (MAE) as the self-supervised feature extractor backbone, using the ViT-Large model pre-trained on ImageNet. To investigate whether self-supervised fine-tuning could enhance the representation quality for label error correction, we fine-tuned the ViT-Large model on the Noisy Ostracods dataset in a self-supervised manner for 100 epochs. Following the AQuA benchmark [56], we evaluated two label correction methods: Area-Under-Margin (AUM) [29] and CINCER [30]. For AUM implementation, we followed the method's theoretical foundation rather than its original implementation. Specifically, we injected 5% of noise from the negative class to each class to construct the AUM calculator following the paper's theoretical statement. In contrast, the paper is constructing a noisy class and moving 5% of data from the clean classes to do AUM calculation. We trained a Resnet-50 model on the injected dataset to acquire the AUM data. As we are more interested in finding the suspicious label in the dataset, our implementation of CINCER is calculating the margin between the predicted label probability and actual label probability to find suspicious samples during inference time. These margins were computed using our best-performing ResNet-50 model. The results of our study are summarized in Table 2.

Table 2: Experiments on different cleaning methods on cleaning of test and validation set. Note the $P.$ is the abbreviation for $Paradoxostomaid$, the majority-wrong classes discussed in Section 3. All numbers are presented as percentages. For SimiFeat methods, entries with mv suffix use majority voting, while others use HOC ranking.

| Method | Hit rate | Feature error hit | Label error hit | Fix prec. | Found pseudo | $P.$ hit rate | Hit rate w/o $P.$ |
|---|---|---|---|---|---|---|---|
| CL | 59.37 | 62.95 | 54.06 | 64.13 | F | 7.94 | 75.41 |
| SimiFeat-CLIP | 26.29 | 30.11 | 20.63 | 17.13 | F | 6.88 | 32.34 |
| SimiFeat-CLIP-mv | 26.29 | 29.05 | 22.19 | 17.29 | F | 7.41 | 32.18 |
| SimiFeat-MAE-trained | 28.18 | 30.32 | 25.00 | 47.16 | T | 3.17 | 35.97 |
| SimiFeat-MAE-trained-mv | 31.07 | 32.21 | 29.38 | 44.67 | T | 2.65 | 39.93 |
| SimiFeat-MAE-raw | 27.55 | 32.21 | 20.63 | 13.83 | T | 12.70 | 32.18 |
| SimiFeat-MAE-raw-mv | 26.79 | 32.00 | 19.06 | 13.87 | T | **14.29** | 30.69 |
| AUM | 29.56 | 24.84 | 36.56 | **86.72** | F | 3.17 | 37.79 |
| CINCER | 53.71 | 56.63 | 49.38 | 54.60 | T | 4.76 | 68.98 |
| NECV | **71.19** | **80.42** | **57.50** | 55.87 | T | 5.82 | **91.58** |

**Results**   All results are evaluated by comparing the noisy labels identified by each method to the manually corrected labels in the test and validation sets. Hit-rate, equivalent to recall, measures the proportion of actual noisy labels identified by the model, making it a crucial metric as it indicates the method's effectiveness in detecting errors. In Table 2, NECV achieved the highest Hit-rate of 71.19%, meaning over 70% of the error samples were detected. Confident Learning (CL) followed with a hit-rate of 59.37%, indicating it found over half of the noisy samples. CINCER found 53.71% of error in the test and validation set. However, all SimiFeat methods performed poorly on this metric, identifying only 26% to 31% of errors. Self-supervised fine-tuning improved the hit-rate by about 5%. AUM had similar performance of 29.56% hit rate.

The hit-rates of feature and label errors, two error types discussed in previous chapters, were also assessed. Feature errors typically require deletion, while label errors require re-labeling. A similar trend to the overall hit-rate was observed, with CL and NECV leading and SimiFeat lagging. Interestingly, SimiFeat with fine-tuned MAE achieved a higher label error hit-rate, suggesting the fine-tuned model may learned distinctive features for ostracods. Expect AUM, all methods performed better at detecting feature errors than label errors, possibly due to distinct feature distributions in corrupted images.

Precision, indicating the ratio of actual noisy samples in the detected list, relates to efficiency—lower precision means more time spent checking clean samples raised as suspecious by the methods. AUM had the highest precision at 86.72%, followed by CL, then NECV. In terms of non-noise sample selection, CL and AUM outperformed NECV. This suggests AUM is trading for precision heavily by low recall (hit rate). For SimiFeat methods, runs with fine-tuned backbones significantly outperformed those without, showing up to a 33% improvement in precision. This suggests that fine-tuning helps SimiFeat select fewer non-noisy samples. Among the genus in the test set, *Cyprideis* was a typo for *Cytherois*. No image should be labeled as *Cyprideis*. If a method included all *Cyprideis* instances in its noise set, it was marked T (True) in the Found Pseudo Class column; otherwise, it was marked F (False). AUM, CL and SimiFeat with CLIP backbones failed to find the pseudo classes.

Methods assuming 'majority is correct', especially those using majority voting, performed poorly on the genus *Paradoxostomid*, where the majority is wrong. The best performer was SimiFeat with the MAE backbone, achieving a 14.29% hit-rate. Interestingly, fine-tuning the backbone significantly dropped SimiFeat's hit rate on *Paradoxostomid*. After removing *Paradoxostomid*, the hit rates increased significantly, with NECV reaching over 90%. This suggests that if clean labels with NECV, excluding *Paradoxostomid*, could remove over 90% of the noise. CL also achieved a 75% hit-rate, with less than 10% of NECV's computational consumptions.

Our initial investigation revealed that the images in the Noisy Ostracods dataset performed poorly when using a k-NN classifier based on features extracted by the CLIP-ViT-L14 and MAE models. This poor performance is due to high $\delta_k$, resulting in inferior $(k, \delta_k)$ label cluster-ability and causing low performance across nearly all metrics [12]. Detailed analyses can be found in the AppendixH.1. The higher performance of SimiFeat with a fine-tuned MAE backbone indirectly supports this,

suggesting that standard pre-trained models are less effective due to the rarity of ostracods in their training datasets. The fine-grained nature of ostracod features adds another layer of difficulty, as the differences between species are subtler than those between cats and dogs often seen in the pretraining datasets. Overall, it may be challenging to clean specialized, fine-grained datasets without training models. Despite being straightforward, NECV proves to be the most reliable method for detecting label errors in fine-grained taxonomic datasets like Noisy Ostracods.

# 5 Limitations

**No Cleaned Training Set** The biggest limitation of the Noisy Ostracods dataset is the absence of a fully cleaned training set. Additionally, cleaned validation and test sets for species are not provided. A cleaned training set would enable researchers to monitor the exact impact of noisy samples on model performance. We are currently working on fully cleaning the dataset to address this issue.

**Taxonomy Is Not Static** Taxonomy is an evolving field, and changes in taxonomic classification can affect the dataset and its results. We will actively follow updates in ostracod taxonomy and update the dataset accordingly. Therefore, for studies using the Noisy Ostracods dataset, it is recommended to keep a record of per-image predictions to accommodate future taxonomic revisions.

# 6 Conclusion and Future Work

We introduced the Noisy Ostracods dataset, a challenging real-world benchmark featuring fine-grained visual differences, class imbalance, and naturally occurring label noise. Our analysis categorized the noise into two practical types: label errors and feature errors, distinguished by their correction requirements. Through comprehensive experiments with current Learning with Noisy Labels (LNL) methods and label correction approaches, we addressed two key questions:

- **Robustness Against Label Noise**: Despite their sophistication, state-of-the-art LNL methods failed to provide significant advantages over standard cross-entropy training with ImageNet pre-training. The baseline cross-entropy model achieved an error rate of 4.02%, below the dataset's noise rate of 5.58%, suggesting that pre-trained models may already possess inherent robustness to real-world noise patterns.
- **Label Correction Effectiveness**: Our experiments revealed that sophisticated label correction methods were consistently outperformed by NECV, a simple ensemble-based cross-validation approach (Algorithm 1). This unexpected result suggests that current methods, while effective on synthetic noise, may not adequately address the complexities of real-world taxonomic datasets with fine-grained features and natural class ambiguities.

These findings highlight the gap between theoretical advances in label noise learning and practical applications in challenging real-world scenarios. They also emphasize the need for developing methods that can handle both fine-grained visual distinctions and natural class ambiguities.

In the next stage, we will clean the training set and provide species-level data. For species-level data, we may provide multi-label annotations per image to address ambiguity. We are continuously capturing new images of the samples, with over 120,000 unlabeled samples collected so far. More unlabeled and machine-labeled samples from around the world are being added to the dataset.

## Acknowledgments and Disclosure of Funding

This research was supported by RGC Research Fellow Scheme of Hong Kong (RFS2223-7S02), Germany/Hong Kong Joint Research Scheme (G-HKU709/21), SKLMP Seed Collaborative Research Fund (SKLMP/SCRF/0031), Seed Funding of the HKU-TCL Joint Research Centre for Artificial Intelligence, and Small Equipment Grant of The University of Hong Kong.

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

# Appendix

## A   Dataset statistics

The dataset is highly imbalanced. The most frequent class, *Sinocytheridea impressa*, comprises over 30% of the entire dataset. The top 19 classes, ranked by frequency in descending order, make up over 90% of the dataset (see Figure 3). Additionally, we included around 1,000 unlabeled data points in the published version of the Noisy Ostracods dataset.

Regarding magnifications, the majority of the images were taken under 50x magnification. To ensure that the ostracods' aspect ratio remained unchanged after resizing to squared images—a common pre-processing step—we added black padding to the images. The average size of the images in the dataset is 476 x 476 pixels.

No potential personally identifiable information or offensive content is included in the dataset.

## B   Data Collection Process

We took the digital images using the Keyence VHX-7000 Microscope [25]. We then cropped the images into smaller sections containing only one grid. Each specimen was further cropped to create the files in the Noisy Ostracods dataset. The process can be viewed in Figure 4. As the figure shows, each grid measures only 3.8 mm, indicating that ostracods are extremely small.

## C   Detailed Feature Errors

The feature error distribution chart, cleaned for the test and validation sets, indicates that 487 samples were identified for deletion. As shown in Figure 5, most errors are categorized as "bad image" and "fragment." The terminology used in the chart corresponds to the raw fixing file provided with the dataset. "Bad image" refers to photo errors, while "fragment" refers to fragmentation errors.

Ostracods have up to nine juvenile stages [1]. The early juvenile stages can be very small, and due to insufficient magnification at 40x and 50x, they might not be clearly visible in the pictures, leading to feature errors being captured. A qualitative showcase of the ostracods' juvenile stages can be seen in Figure 6. The term "reverse position" refers to position errors, as illustrated in Figure 2. At this stage, we decided to delete these samples. However, they could be useful for constructing 3D models of ostracods since they are simply images of ostracods viewed from different angles.

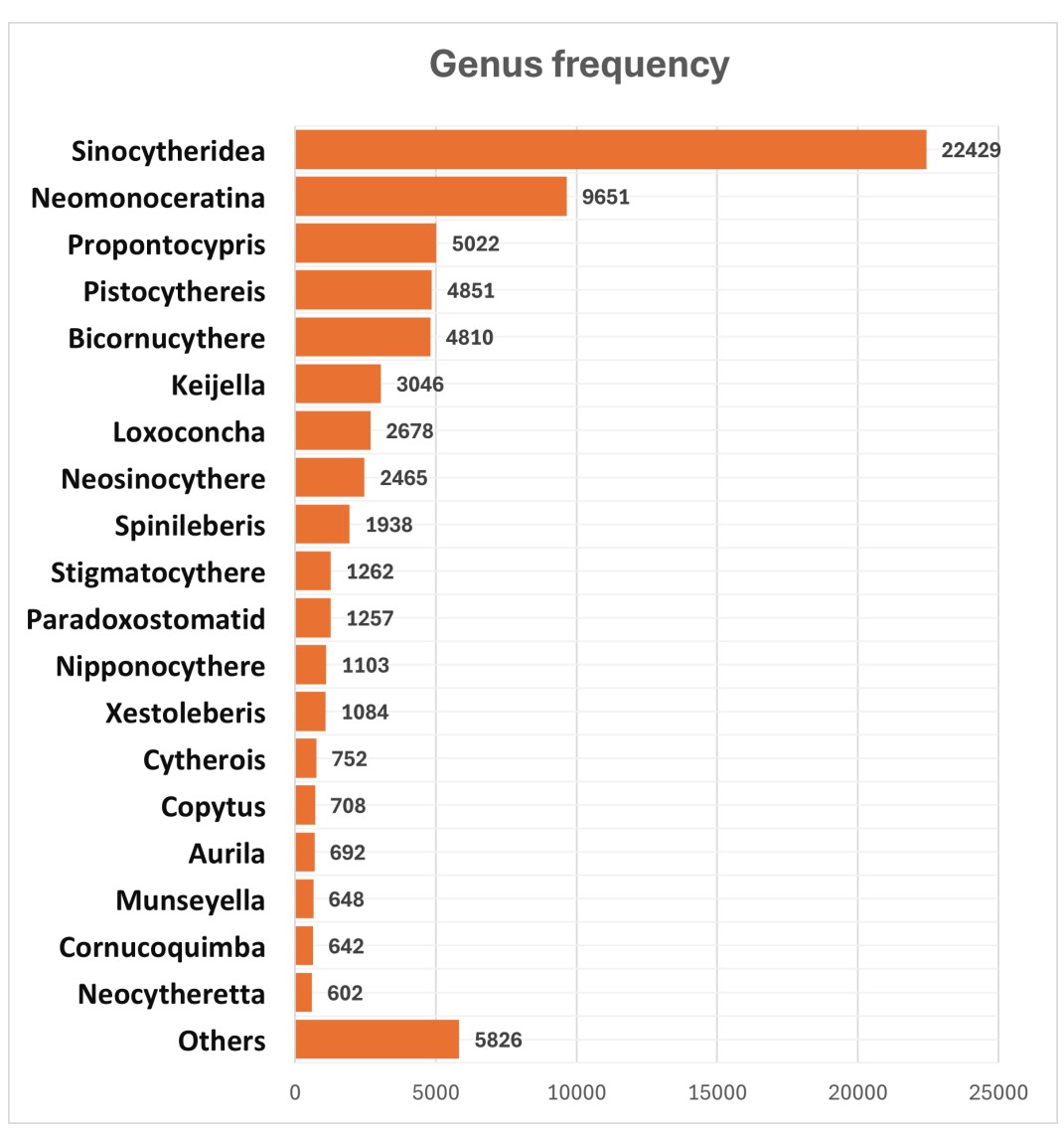

Figure 3: The genus frequency distribution for the dataset.

## D Transition matrices

The transition matrices are organized with rows representing the ground truth labels after curation and columns representing the original labels. Notably, for the pseudo class *cyprdeis*, the number of ground truth images should be 0. Consequently, the row corresponding to *cyprdeis* is is empty (see Figures 9, 8, and 7), indicating that no ground truth should be labeled as *cyprdeis*. Additionally, the negative class has 0 positive samples in both the test and validation sets. All samples in the negative class are feature error samples that have been flipped from other classes.

## E Noise transition matrix based methods

Transition matrices are widely used to model label noise patterns in learning with noisy labels [41, 42, 43]. This approach models label errors as probabilistic transitions between classes through a transition matrix $T(x)$. For class-dependent label noise, the transition process can be formulated as:

$$P(\tilde{y}_i = e|x_i) = \sum_g P(\tilde{y}_i = e|y^* = g)P(y^* = g|x_i) = \sum_g T^*_{eg}P(y^* = g|x_i) \tag{4}$$

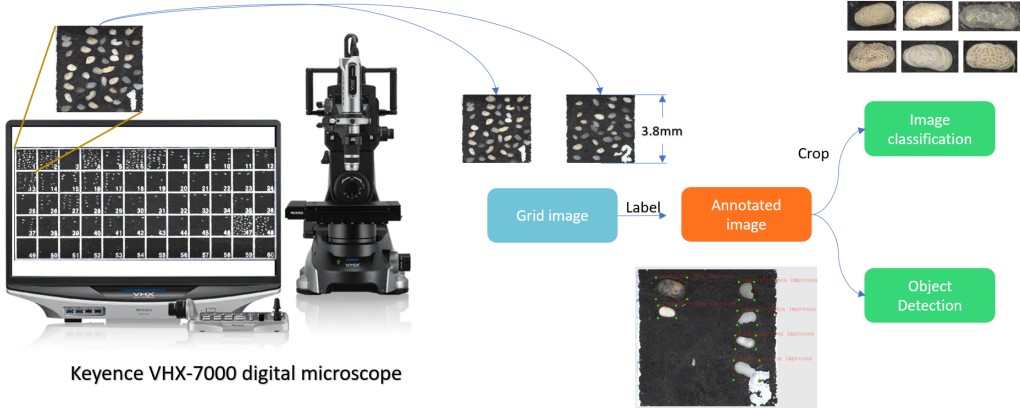

Figure 4: The collection process.

where $T_{eg}^*$ represents the probability of a true label $g$ being observed as label $e$. Based on this formulation, we implemented the loss function following [27]:

$$\mathcal{L}(\theta, T) = -\frac{1}{N} \sum_{n=1}^{N} \log \tilde{p}(\tilde{y} = \tilde{y}_n | \mathbf{x}_n, \theta, T) = -\frac{1}{N} \sum_{n=1}^{N} \log \left( \sum_i T_{\tilde{y}_n i} p(y^* = i | \mathbf{x}_n, \theta) \right) \tag{5}$$

where $\theta$ represents the model parameters. Unlike traditional approaches that estimate the transition matrix, we utilized the ground truth transition matrix derived from our manual validation process (Figure 9). This approach offers two advantages:

- It eliminates transition matrix estimation errors
- It provides an upper bound for transition matrix-based methods, assuming class-dependent noise holds

However, two important caveats should be noted:

- Using the ground truth transition matrix introduces information leakage, making the results primarily theoretical
- The method's poor performance, despite using the true transition matrix, suggests that the class-dependent noise assumption may not adequately capture the noise patterns in the Noisy Ostracods dataset

This analysis indicates that real-world fine-grained classification tasks may exhibit more complex noise patterns than the traditional class-dependent noise model can represent.

## F    Hyper-parameters and training details

In our study, we employed the following training parameters to optimize our model: The models were trained with a dynamic relationship between the batch size $k$ and the learning rate, defined as $l \times k$, following the approach outlined in [61]. For ResNet 50, the scaling factor $l$ was set to 0.0001, while for ViT (Vision Transformer), $l$ was set to 0.00008. This relationship ensures that the learning rate scales linearly with the batch size, allowing for stable and efficient training. We utilized the SGD (Stochastic Gradient Descent) optimizer with a momentum of 0.9. The momentum term helps accelerate gradient vectors in the right direction, thus leading to faster convergence. SGD updates the parameters iteratively based on the gradient of the loss function with respect to the parameters. The momentum parameter in the SGD optimizer helps in smoothing oscillations and speeds up convergence by taking into account previous gradients. It was set to 0.9. We adapted the StepLR (Step Learning Rate) scheduler. This scheduler decays the learning rate by a factor of $\gamma$ every step_size epochs. This helps in fine-tuning the learning rate as training progresses, which can lead to better and more stable convergence. The

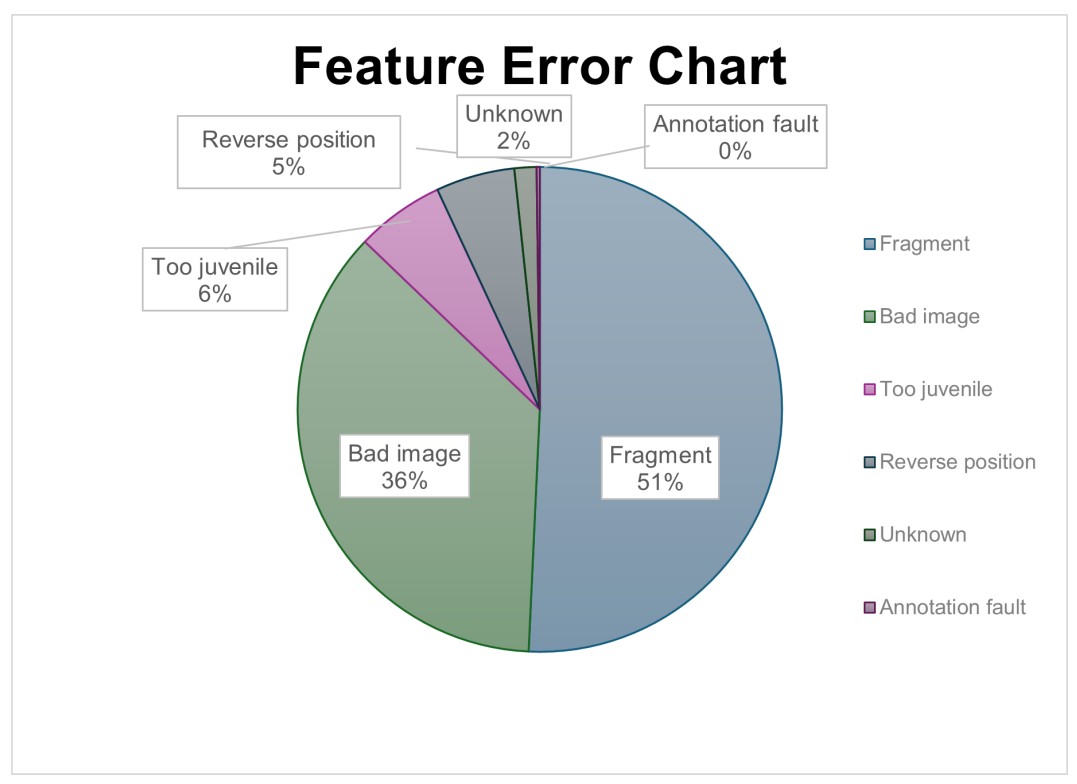

Figure 5: The detailed feature errors. The wording is slightly different from the article.

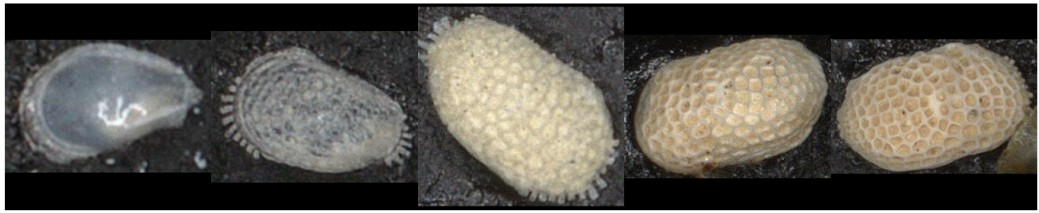

Figure 6: A qualitative showcase of the juvenile to adult stages of *Pistocythereis bradyi*. From left to right is in juvenile to adult order.

step_size was set to 7, meaning that every 7 epochs, the learning rate is adjusted. The decay factor $\gamma$ was set to 0.1, meaning that at each step, the learning rate is multiplied by 0.1, effectively reducing it by an order of magnitude.

For co-teaching, co-teaching plus, and loss clipping, the forget rate is set to 0.05 with 30 warming up rounds. For Divide-Mix, we did not change any official hyperparameters except the sampled batch number was reduced to 218 instead of 1000. For SimiFeat, we adapted the hyperparameters directly from the Docta [62] implementation, adjusting the sample size to 0.7 times our dataset size. For MAE fine-tuning, we used the official implementation's hyperparameters. We performed a CNN-style average pooling on the last layer output of the encoder part, treating it as 1024 channels of $16 \times 16$ features and doing channel-wise pooling. Seeds is 0.

# G   Project-specific Error (Privileged Information)

Project-specific error is highly dependent on the project, as certain types of errors, such as typos, may only appear in specific projects. This kind of information is decried as privileged information and widely used in LNL[63]. In the Noisy Ostracods dataset, a sample file might look like:

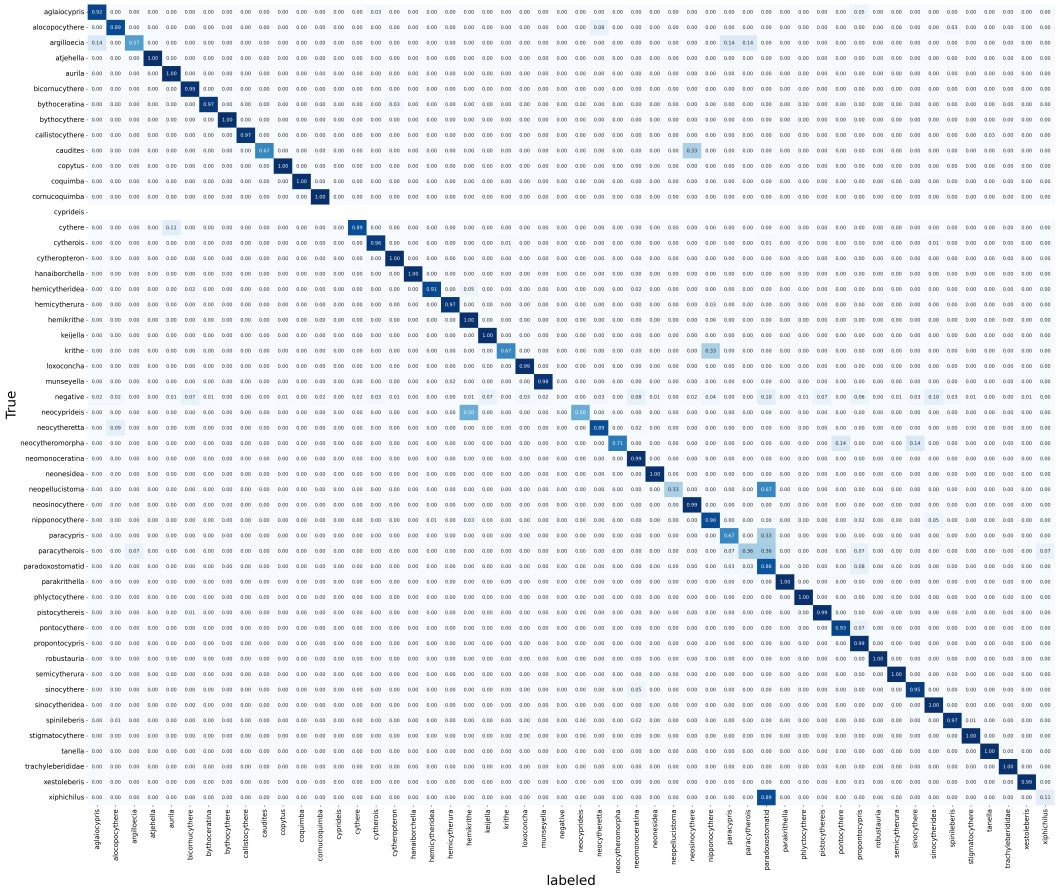

Figure 7: The transition matrix at genus level for validation and test set combined.

```
/[Any data path]/keijella kloempritensis/HK14DB1C_80_81_4_ind8.tif
```

Here, *Keijella kloempritensis* refers to the species in the image. The project is identified as the first component split by an underscore ('_'). In the shown file name, the project is HK14DB1C.

An example of a project-specific error is that project HKUV12 used "$cf.$" while others used "$aff.$" for *Callistocythere reticulata*. From these observations, we argue that this type of error is a special form of annotator-specific error. As shown in Table 3, even though all labels were provided by the same expert, the expert's performance varied among projects, leading to mistakes specific to certain projects. The highest error rates were found in projects HK14DB1C and HKUV12, both with over 8% error. The lowest error rate was in project HK14TLH1C, with around 1% error. Given the high variance in error rates between projects, we suggest that project-specific error could be considered a special case of annotator-specific error, given that the same annotator exhibits such high variance.

## H    Interval included table

**Dynamic Loss Implementation**    We implemented Dynamic Loss [64] with several adaptations for our dataset characteristics. While this method typically combines meta-learning and adaptive sampling to address both label noise and class imbalance, we made two significant modifications to the original implementation:

First, we simplified the hierarchical sampling module. The original method ranks samples by loss values and assigns them to 10 bins, sampling low-loss samples from higher bins for the meta set. However, this approach proved unsuitable for the Noisy Ostracods dataset, where many classes have fewer than 10 samples, making bin-based sampling impractical. The author's did not include the hierarchical sampling module in their Imagenet-LT experiments. Imagenet-LT[65] has serval classes with less than 5 images which may be the reason

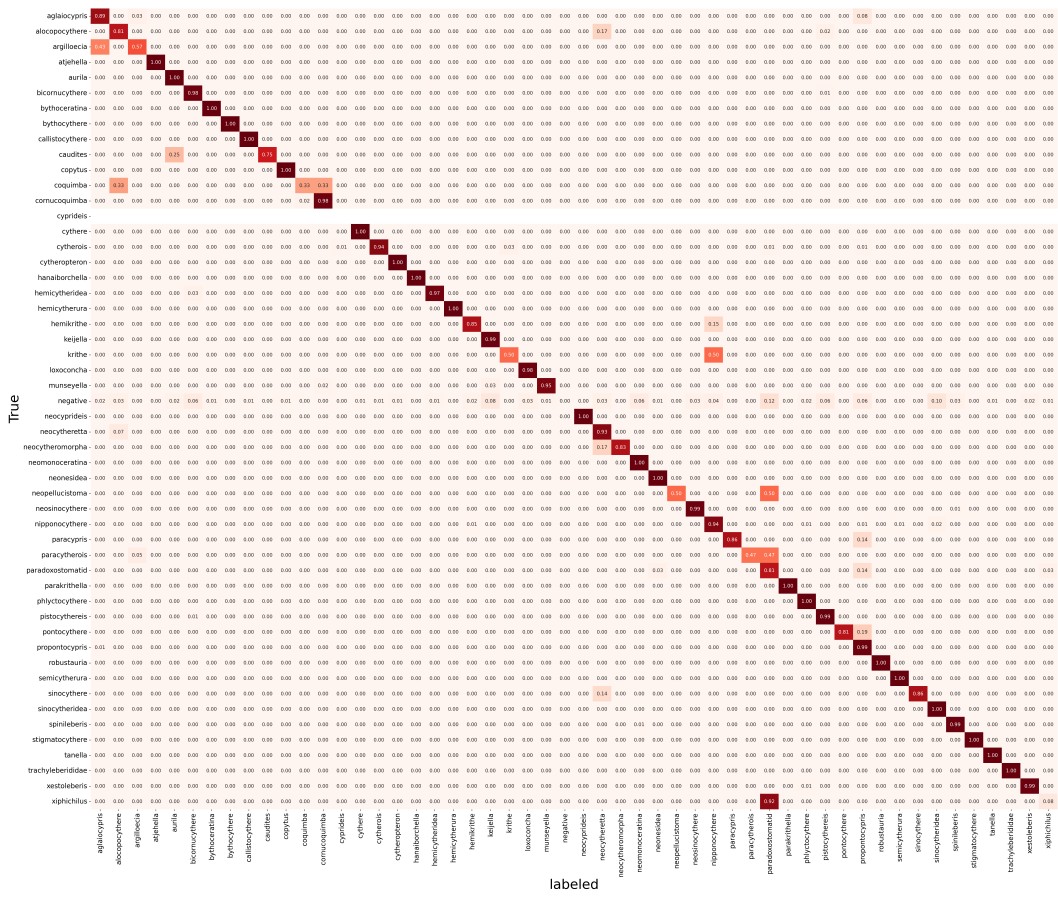

Figure 8: The transition matrix at genus level for test set

Table 3: The project-specific error tracking

| Project | Error rate | Annotation year | Annotator | Validator |
|---------|-----------|-----------------|-----------|-----------|
| HK14DB1C | 8.07% | 2015 | C | H |
| HK14DB2C | 6.87% | 2015 | C | H |
| HK14PCR1C | 2.85% | 2015 | C | H |
| HK14TLH1C | 1.21% | 2015 | C | H |
| HK14TLH2C | 4.07% | 2015 | C | H |
| HKUV12 | 8.31% | 2014 | C | H |
| rawSample | 3.52% | 2022 | C | N/A |
| subSurface | 3.81% | 2014 | C | H |
| surface | 4.54% | 2014 | C | H |

that the author did not including the module. As a result, we used the cleaned validation set directly as the meta set for learning class-specific margins in the softmax function.

Second, while the method performed well with ResNet-50, we encountered technical limitations with ViT-B-16. The unavailability of second-order derivatives for certain ViT modules prevented successful training with this architecture. Consequently, we excluded the ViT results from our main comparison.

It is important to note that our implementation's strong performance may be partially attributed to information leakage from using the cleaned validation set as the meta set. This modification, while necessary for our dataset's characteristics, deviates from the original method's more generalizable approach.

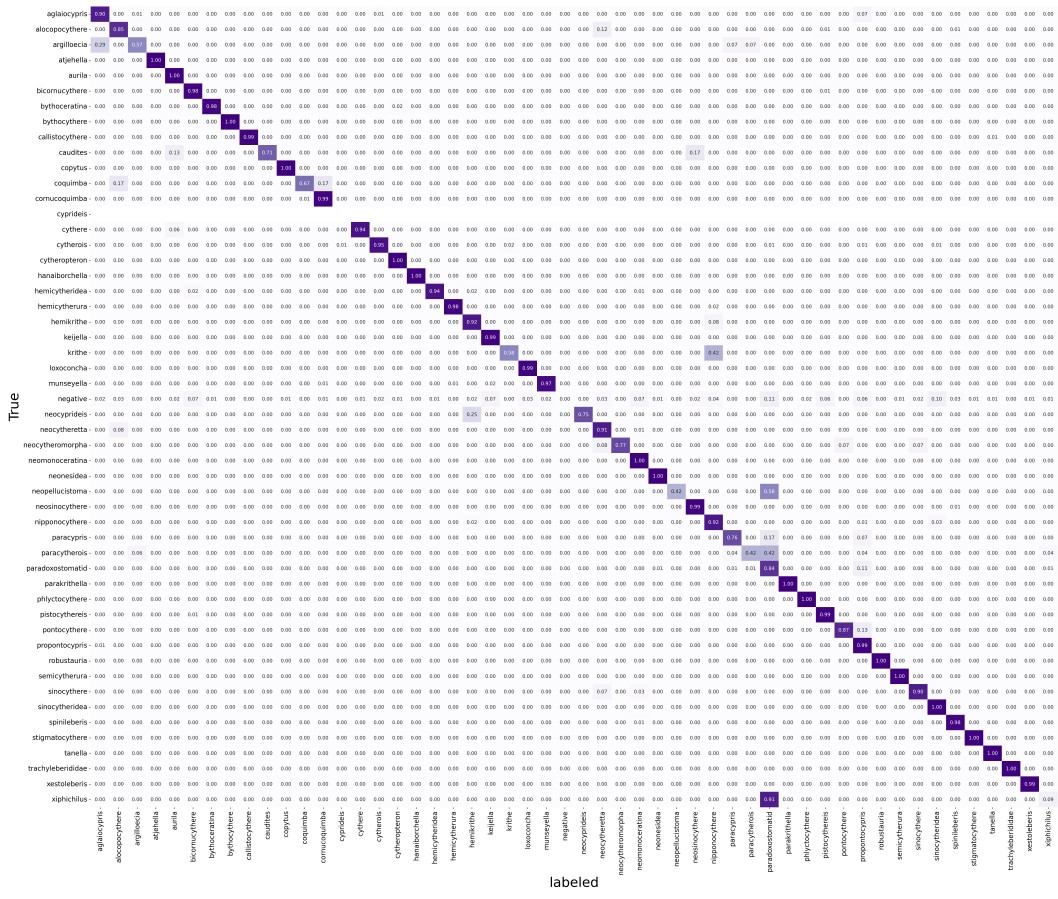

Figure 9: The transition matrix at genus level for validation and test set

Table 4: Experiments with ResNet-50. The average and variance of the metrics over 5 runs are reported. Acc: Accuracy; P@: Precision; R@: Recall; F1@: F1 Score.

| Method | Acc | P@ | R@ | F1@ |
|---|---|---|---|---|
| CE | $95.77 \pm 0.35$ | $83.31 \pm 4.02$ | $75.20 \pm 2.78$ | $76.51 \pm 2.99$ |
| Mixup+Cutmix | $92.62 \pm 3.62$ | $64.12 \pm 12.27$ | $56.26 \pm 11.42$ | $56.31 \pm 11.51$ |
| CL | $94.93 \pm 0.34$ | $78.62 \pm 3.78$ | $70.38 \pm 5.23$ | $72.34 \pm 4.43$ |
| Loss-clip | $62.43 \pm 32.98$ | $37.39 \pm 34.82$ | $34.82 \pm 33.08$ | $34.82 \pm 33.49$ |
| Co-teaching | $94.79 \pm 0.01$ | $74.19 \pm 1.03$ | $66.69 \pm 0.97$ | $68.19 \pm 0.83$ |
| Co-teaching+ | $94.27 \pm 0.40$ | $77.01 \pm 3.99$ | $68.47 \pm 2.33$ | $69.97 \pm 2.27$ |
| DivideMix | $93.13 \pm 2.00$ | $46.78 \pm 7.04$ | $49.63 \pm 6.96$ | $47.81 \pm 6.94$ |
| SAM | $94.49 \pm 0.29$ | $77.34 \pm 1.69$ | $69.70 \pm 2.16$ | $70.55 \pm 1.89$ |
| PLM | $96.70 \pm 0.06$ | $79.22 \pm 1.75$ | $72.81 \pm 1.81$ | $74.09 \pm 1.65$ |
| SDN | $93.91 \pm 1.13$ | $75.92 \pm 5.02$ | $69.33 \pm 2.55$ | $69.80 \pm 3.60$ |
| LW | $95.21 \pm 0.23$ | $84.23 \pm 2.78$ | $73.66 \pm 2.15$ | $75.29 \pm 2.34$ |
| Margin | $96.11 \pm 0.04$ | $80.24 \pm 2.11$ | $73.63 \pm 1.59$ | $74.49 \pm 1.51$ |
| Transition* | $82.62 \pm 1.65$ | $60.23 \pm 3.29$ | $52.30 \pm 3.06$ | $53.53 \pm 2.94$ |

* Transition matrix is trained on a subset of 51 classes with more than 10 samples each.

Table 5: Experiments with ViT-B-16. The average and variance of the metrics over 5 runs are reported. Acc: Accuracy; P@: Precision; R@: Recall; F1@: F1 Score.

| Method | Acc | P@ | R@ | F1@ |
|---|---|---|---|---|
| CE | $94.21 \pm 1.59$ | $78.17 \pm 6.95$ | $70.13 \pm 6.96$ | $71.58 \pm 6.73$ |
| Mixup+Cutmix | $89.27 \pm 0.62$ | $55.40 \pm 5.29$ | $47.54 \pm 4.18$ | $48.52 \pm 4.64$ |
| CL | $82.20 \pm 21.08$ | $34.37 \pm 23.16$ | $30.38 \pm 22.37$ | $30.73 \pm 22.54$ |
| Loss-clip | $93.17 \pm 1.86$ | $44.80 \pm 6.95$ | $50.05 \pm 5.37$ | $46.76 \pm 6.27$ |
| Co-teaching | $94.96 \pm 1.54$ | $80.77 \pm 2.96$ | $71.59 \pm 3.39$ | $73.07 \pm 3.16$ |
| Co-teaching+ | $95.22 \pm 1.07$ | $74.66 \pm 5.50$ | $68.89 \pm 5.24$ | $69.47 \pm 5.32$ |
| DivideMix | $83.18 \pm 1.40$ | $23.81 \pm 5.60$ | $22.66 \pm 6.03$ | $19.68 \pm 6.29$ |
| SAM | $93.10 \pm 0.32$ | $74.86 \pm 3.72$ | $64.00 \pm 2.48$ | $66.28 \pm 2.34$ |
| PLM | $96.10 \pm 0.37$ | $81.04 \pm 3.09$ | $73.62 \pm 2.67$ | $75.12 \pm 2.59$ |
| SDN | $87.75 \pm 4.92$ | $70.75 \pm 4.14$ | $56.82 \pm 4.02$ | $59.29 \pm 4.34$ |
| LW | $92.33 \pm 2.83$ | $72.40 \pm 9.03$ | $63.87 \pm 9.04$ | $65.17 \pm 9.25$ |
| Margin[†] | – | – | – | – |
| Transition* | $82.89 \pm 0.20$ | $57.87 \pm 2.44$ | $50.32 \pm 1.45$ | $51.16 \pm 1.57$ |

*Transition matrix is trained on a subset of 51 classes with more than 10 samples each.
† Results unavailable due to incompatibility of second-order derivatives with ViT architecture.

## H.1 Supporting data for SimiFeat analyses

In the main article, we did analyses of bad performance of SimiFeat methods without using number to support the statements. The KNN classifier accuracy using embedding from MAE[60] models are given in table 6. As shown in the table, as k approaches infinity, the accuracy of k-nn classifier just approaches around 31%. We observed that 31% is near the percentage of the most majority class *Sinocytheridea impressa*'s composition in the dataset. *Sinocytheridea impressa* composed of 31.38% of images in the dataset. As a result, we explain the bad performance of SimiFeat as low clusterability of the dataset features extracted from MAE and CLIP.

Table 6: Accuracy comparison of KNN classifiers from finetuned and pretrained MAE across different NN counts.

| NN count | Finetuned Acc. | Pretrained Acc. |
|---|---|---|
| 5 | 12.3054% | 13.7873% |
| 7 | 16.0557% | 9.7426% |
| 11 | 30.1471% | 30.5106% |
| 13 | 30.5286% | 30.9405% |
| 17 | 30.8797% | 30.9723% |
| 19 | 30.9074% | 30.9806% |
| 23 | 30.9295% | 30.9972% |
| 29 | 30.9488% | 31.0041% |
| 31 | 30.9530% | 31.0041% |

# I   Licenses

Table 7: Licenses of Various Software

| Software | License |
|---|---|
| Docta.AI | NonCommercial |
| cleanlab | AGPL-3.0 License |
| Pytorch | BSD 3-Clause License |

# J   Compute resources

To produce the results, 3 Nvidia RTX 4090 GPUs, 2 Nvidia RTX 4090 Laptop GPUs and one Nvidia RTX 3090 GPU were used.

