# Documentation for The Noisy Ostracods Dataset

**Jiamian Hu[1], Yuanyuan Hong[1], Yihua Chen[1,2], He Wang[1,3], Moriaki Yasuhara[1]**

[1]The University of Hong Kong
[2]The University of Tokyo
[3]Nanjing Institute of Geology and Palaeontology
{jiamianh, u3001143, yihuaac, hw0701}@connect.hku.hk, yasuhara@hku.hk

## 1 Motivation and Backgrounds

The *Noisy Ostracods* dataset is a real-world taxonomy dataset characterized by various types of noise. It was created out of the need for a clean taxonomy dataset and the challenges we encountered during the cleaning process in our real use case. Our goal was to provide a benchmark for evaluating the performance of robust machine learning methods and label correction algorithms from a practical perspective. The imbalanced and fine-grained nature of the dataset introduces additional challenges to these methods.

The document is made by adapting the most relevant questions from datasheets for datasets[1] according to the property of our datasets.

## 2 Data Collection Detail

The dataset included Ostracods from the Hong Kong marine sediments collected over the past 10 years. The goal for collecting the ostracods is to exploring the quantitative correlation between common ostracod species composition and environmental factors[2, 3]. The details of collection process of sediments are available in the original works[3, 2, 4]. This document primarily focus on the collection process of the photos and the effort we made to ensure the quality of the dataset.

### 2.1 Collection process of Noisy Ostracods 2022

Ostracod samples from the original studies are stored in standard 60-grid microfossil slides. These slides are photographed using a Keyence-VHX-7000 microscope [5]. Images are captured at magnifications ranging from 40X to 80X. A sample slide image is shown in Figure 1. Initially, all images were taken at 50X magnification. In most cases, the photos resemble the surface_SS6 slide in Table 1. However, when the image resolution is excessively high, the microscope automatically compresses the images. As illustrated in Table 1, for slide HK14TLH1C_151_152, the expected resolution at 50X magnification is approximately 23000*9600. The actual resolution, however, is 11599*4841, roughly half of the anticipated resolution. We conducted experiments on slide HK14TLH1C_136_137 to determine the optimal resolution. Even after compression, the 80X images were still larger in native resolution than the 40X images. However, capturing photos at 80X took four times longer, with no significant improvement in quality. Consequently, we decided to use 40X magnification for slides where 50X images were compressed. Following the preparation sequence shown in Figure 1, the slides are cropped into grids for easier processing. A typical 50X grid is sized around 1600*1600 pixels. We then annotate the ostracods from the grid images using `labelImg` [6]. Since the original taxonomy record is organized per grid, we simply join the record with the annotation by grid number. Before this joining process, we cleaned the typographical errors in the original identification file. The typos addressed here were different from those present in the final version of the *Noisy Ostracods* dataset. Primarily, we corrected misspellings of genus and species names, such as changing *Xestol-*

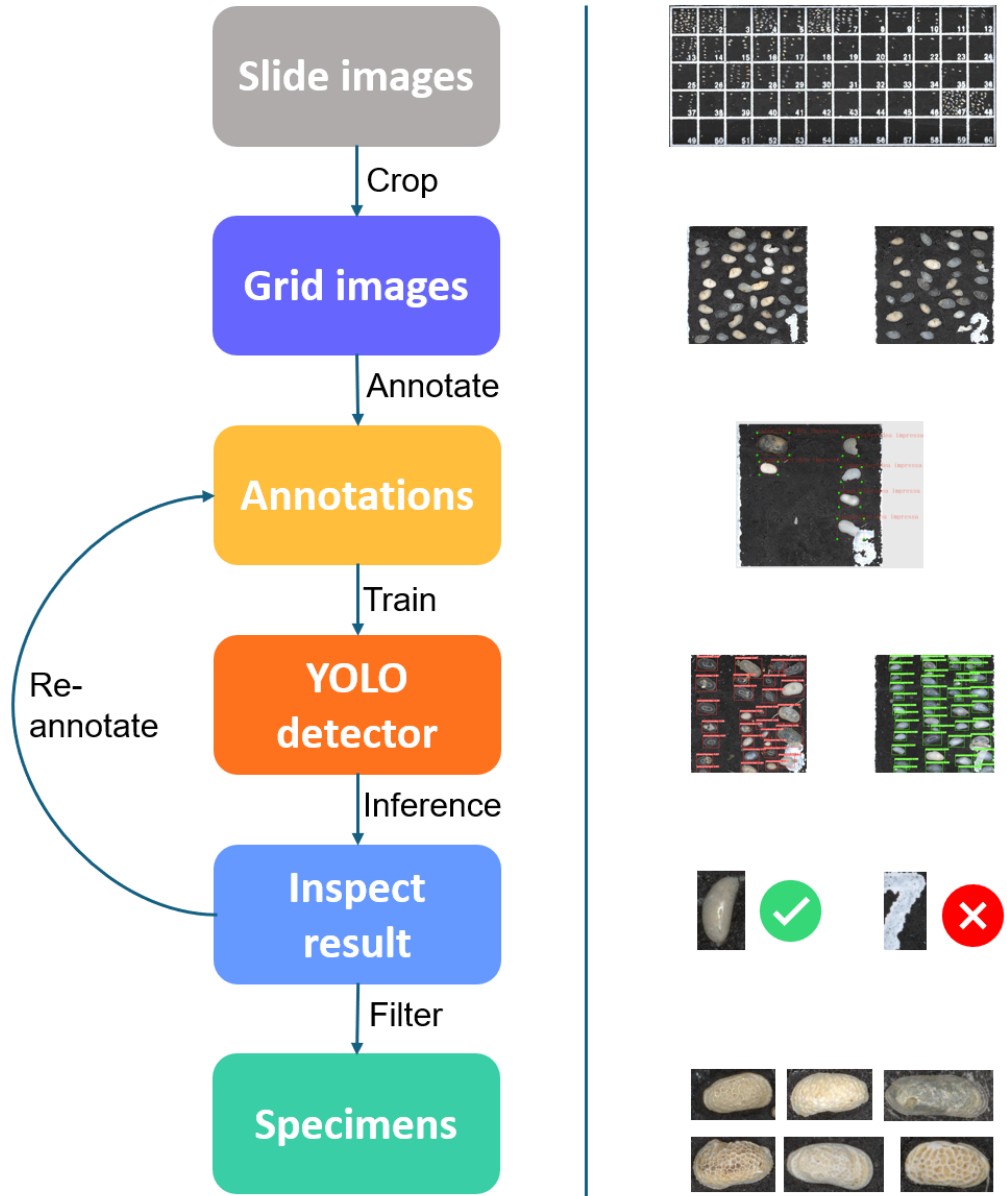

Figure 1: Illustrated collection process of the noisy ostracods dataset. Left column: simplified flow chart. Right column: illustrated examples.

beris to *Xestoleberis*. However, semantic typos, such as misnaming *Cytherois* as *Cyprideis*, could not be detected at this stage because both genera are valid in ostracod taxonomy literature; such errors can only be identified through precise specimen checks.

We employ an iterative approach for annotation: we start by manually annotating around 180 grids and then train the initial detector using YOLO [7]. Using this new YOLO model, we annotate an additional 180 grids, correcting any errors in the bounding boxes. We then train the next model using all 360 annotated grids. This process is repeated until we have a training set of 6000 grids. The model trained on these 6000 grids is then used to annotate subsequent images, which are manually checked to eliminate errors. At this stage, we crop out the specimens to create an ostracod taxonomy dataset. However, the amount of error in the dataset proved to be non-negligible, necessitating multiple revisions. We retained the initial version of the dataset, making it available as *Noisy Ostracods 2022*.

Table 1: Resolution data for slides at different magnifications. The images with red actual resolutions are being compressed.

| Slide | Magnification | Expected resolution | Actual resolution |
|---|---|---|---|
| HK14TLH1C_151_152 | 40X | 18548*7764 | 18548*7764 |
| HK14TLH1C_151_152 | 50X | 23185*9705 | 11711*4929 |
| HK14TLH1C_136_137 | 40X | 18541*7721 | 18541*7721 |
| HK14TLH1C_136_137 | 50X | 23176*9651 | 11599*4841 |
| HK14TLH1C_136_137 | 80X | 37082*15422 | 18628*7759 |
| surface_SS6 | 50X | 21525*9402 | 21525*9402 |

## 2.2 Building the Noisy Dataset

When building the *Noisy Ostracods* dataset, we initially aimed to completely eliminate the errors present in the *Noisy Ostracods 2022* version. We identified several issues in the *Noisy Ostracods 2022* dataset:

- **Including non-ostracods**: Due to failures in the YOLO detector, some slides included non-ostracods. An example of this error, found in the *Xestoleberis* genus, is illustrated in Figure 1.

- **Missing ostracods**: Some ostracods were not detected, also due to failures in the YOLO detector.

- **Bad images**: Some slides photographed at an early stage were too bright, as shown in the left image of Figure 2. This issue was caused by incorrect camera settings.

- **Dual records in grids**: Ideally, each grid should contain only one species of ostracods. However, some grids had more than one species recorded due to practical reasons, such as the mixture of hard-to-distinguish species in the same grid. In creating the 2022 version, we skipped all such records.

Based on the errors identified in the 2022 version, we applied the following fixes:

1. Checked the annotation file: Re-annotated ostracods with incorrect bounding boxes and removed non-ostracods. Also annotated ostracods missed by the YOLO detector.

2. Checked the image files: Retook images that were too bright.

3. Consulted experts: Asked the experts who provided the original identifications to manually annotate the respective species on the images.

After applying these fixes, the majority of errors caused by the YOLO detector and camera settings were resolved. The number of YOLO detector failure found in current cleaning process on the *Noisy Ostracods* dataset is less than 20. However, unlike the thorough cleaning described in the main article, this round of inspection was conducted by non-experts. This led to some remaining errors, such as fragmentation errors and semantic typos, as detailed in the main article. Nevertheless, we believe that the errors in the current version of the *Noisy Ostracods* dataset have been minimized from a procedural perspective. The remaining noise primarily consists of hard-to-fix errors, such as expert misidentifications. Consequently, this version can serve as a valuable resource for researchers studying real-world noise and its effects on the performance of machine learning models, particularly in the context of trustworthy machine learning.

## 3 Dataset composition

## 3.1 Images

The *Noisy Ostracods* dataset comprises 71,466 images, while the *Noisy Ostracods 2022* dataset contains 68,458 images. The difference in image count is primarily due to the addition of specimens from grids with mixed records. The image files are organized by their *annotated* species. If a

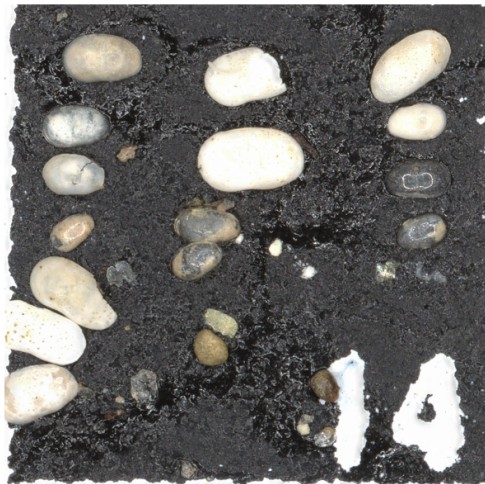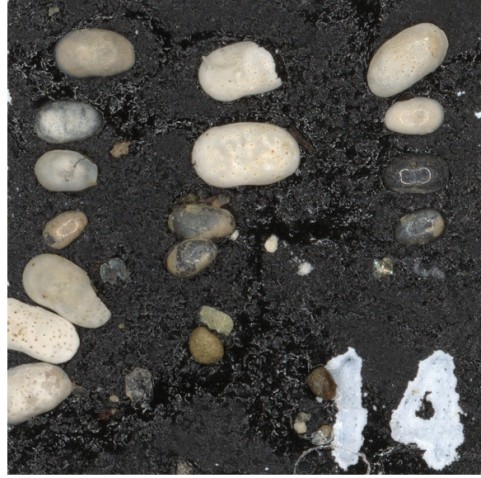

Figure 2: Illustration of camera parameter failure error. Left: a grid image that is too bright. Right: the re-took image.

specimen's species is unconfirmed in the original identification, it is moved to the genus class *genus unidentified*. A concrete example of the file structure of the dataset is shown in the file tree below.

```
Root Folder
└── Noisy_ostracods
    ├── sinocythere sinensis
    │   └── images
    ├── sinocythere unidentified
    │   └── images
    └── other species
        └── images
```

In the file tree, the *Sinocythere unidentified* included the specimens of *Sinocythere* with unconfirmed species. All versions of the dataset adhere to this structure. To preserve the best quality of the images, all RGB images are stored in uncompressed `.tif` format. In addition to ostracod images, we included 9,117 negative samples by randomly cropping backgrounds that do not belong to ostracod bounding boxes and by adding random photos from the Endless Forams dataset [8] as a negative class. We also have 876 images without labels, as the original records for these grids marked their genus and species with a `?` (question mark). Some of these can be identified to existing genera and species. In total, the *Noisy Ostracods* dataset contains 81,459 labeled images.

## 3.2 Labels

The labels are stored in `.csv` files to enable flexibility. Each record in the label file includes two entries: the image path and the label number. For example, a row in the label file might look like `alocopocythere goujoni/HKUV12_465_467_38_ind6.tif,2`. This line indicates that the file `HKUV12_465_467_38_ind6.tif` located in the `alocopocythere goujoni` folder has an image label of 2. Each label file is accompanied by a corresponding guidance file that maps the image label number to its genus/species name in string format. These guidance files are `.txt` files that list the genus/species names, with the line number corresponding to the label number in the label files.

This structure provides the flexibility to update the genus/species of the images by simply changing the image label number in the corresponding label file whenever an error is detected. If a typographical or semantic error is found, we can correct it by updating the corresponding guidance and label files. Such a structure ensures that any necessary adjustments can be made efficiently, maintaining the accuracy and integrity of the dataset.

The provided label files and label guidance files are:

- **Noisy label files**: `ostracods_genus_final_train.csv`, `ostracods_genus_final_val.csv`, `ostracods_genus_final_test.csv`, `ostracods_species_final_train.csv`, `ostracods_species_final_val.csv`, and `ostracods_species_final_test.csv`. These files contain the training, validation, and test splits for the *Noisy Ostracods* dataset at the genus and species levels.
- **Clean label files**: `ostracods_genus_clean_test.csv` and `ostracods_genus_clean_val.csv`. These files contain the cleaned test and validation splits at the genus level.
- **Guidance files**: `ostracods_genus_final_guide.txt` and `ostracods_species_final_guide.txt`. The guidance files include 139 rows for species and 79 rows for genus. The same guidance is used for both noisy and clean data.

The clean label files are slightly smaller than their noisy counterparts because some noisy files have been deleted. At the current stage, we are beginning the comprehensive cleaning of the dataset at the genus level and have inspected some problematic species. The known issues are listed in Table 2.

As shown in the table, the inconsistent usage of *Confer* (cf.) and *Affinis* (aff.) across different projects has introduced many pseudo-classes. Additionally, a typo of *Spinileberis quadriaculeata* was introduced during the re-annotation of grids containing multiple species. Resolving issues with some challenging species, such as *Pistocythereis subovata* and *Pistocythereis bradyi*, may result in multi-class labels, as it is nearly impossible to distinguish these species from a single image.

Table 2: Full list of genus and species in the *Noisy Ostracods* dataset. Known problems are listed.

| *Genus* | *Species* | Count | **Known problems** |
|---|---|---|---|
| *aglaiocypris* | - | 426 | - |
| *alataconcha* | *alataconcha cf. pterogona* | 8 | - |
| *alataconcha* | - | 1 | - |
| *alocopocythere* | *alocopocythere goujoni* | 285 | - |
| *alocopocythere* | *alocopocythere kendengensis* | 67 | - |
| *alocopocythere* | *alocopocythere profusa* | 108 | - |
| *alocopocythere* | - | 5 | - |
| *argilloecia* | *argilloecia lunata* | 23 | - |
| *argilloecia* | - | 29 | - |
| *atjehella* | *atjehella cf. semiplicata* | 68 | Possibile duplicate with *atjehella semiplicata* |
| *atjehella* | *atjehella kingmai* | 4 | - |
| *atjehella* | *atjehella semiplicata* | 2 | Possibile duplicate with *atjehella cf. semiplicata* |
| *aurila* | *aurila aff. corniculata* | 11 | Possibile duplicate with *aurila cf. corniculata* |
| *aurila* | *aurila cf. corniculata* | 4 | Possibile duplicate with *aurila aff. corniculata* |
| *aurila* | *aurila cf. disparata* | 245 | Possibile merge with *aurila disparata* |
| *aurila* | *aurila cf. hataii* | 83 | - |
| *aurila* | *aurila cf. uranouchiensis* | 5 | - |
| *aurila* | *aurila disparata* | 30 | Possibile merge with *aurila cf. disparata* |
| *aurila* | - | 314 | - |
| *bicornucythere* | *bicornucythere bisanensis* | 3177 | - |
| *bicornucythere* | - | 1633 | - |
| *bythoceratina* | *bythoceratina callidictya* | 1 | Possibile merge with *bythoceratina cf. callidictya* |
| *bythoceratina* | *bythoceratina cassidoidea* | 13 | - |
| *bythoceratina* | *bythoceratina cf. angulata* | 1 | - |
| *bythoceratina* | *bythoceratina cf. callidictya* | 21 | Possibile merge with *bythoceratina callidictya* |

| Genus | Species | Count | Known problems |
|---|---|---|---|
| *bythoceratina* | *bythoceratina cf. orientalis* | 11 | - |
| *bythoceratina* | *bythoceratina cf. robusta* | 2 | Possibile merge with *bythoceratina robusta* |
| *bythoceratina* | *bythoceratina orientalis* | 4 | - |
| *bythoceratina* | *bythoceratina robusta* | 9 | Possibile merge with *bythoceratina cf. robusta* |
| *bythoceratina* | *bythoceratina sheyangensis* | 32 | - |
| *bythoceratina* | - | 266 | - |
| *bythocypris* | - | 7 | - |
| *bythocythere* | - | 20 | - |
| *callistocythere* | *callistocythere aff. reticulata* | 9 | Possibile duplicate with *callistocythere cf. reticulata* |
| *callistocythere* | *callistocythere aff. undulatifacialis* | 263 | Possibile merge with *callistocythere undulatifacialis* |
| *callistocythere* | *callistocythere asiatica* | 16 | - |
| *callistocythere* | *callistocythere cf. multirugosa* | 19 | - |
| *callistocythere* | *callistocythere cf. nipponica* | 7 | - |
| *callistocythere* | *callistocythere cf. reticulata* | 2 | Possibile duplicate with *callistocythere aff. reticulata* |
| *callistocythere* | *callistocythere undata* | 5 | - |
| *callistocythere* | *callistocythere undulatifacialis* | 10 | Possibile merge with *callistocythere aff. undulatifacialis* |
| *callistocythere* | - | 45 | - |
| *cathacythere* | *cathacythere reticulata* | 6 | - |
| *caudites* | - | 39 | - |
| *cletocythereis* | - | 3 | - |
| *copytus* | *copytus posterosulcus* | 708 | - |
| *coquimba* | *coquimba cf. ishizakii* | 12 | - |
| *coquimba* | - | 10 | - |
| *cornucoquimba* | *cornucoquimba cf. gibboidea* | 491 | - |
| *cornucoquimba* | *cornucoquimba leizhouensis* | 77 | - |
| *cornucoquimba* | *cornucoquimba pustulata* | 5 | - |
| *cornucoquimba* | - | 69 | - |
| *cyprideis* | - | 12 | According to the photos, should be *cytherois* |
| *cythere* | *cythere omotenipponica* | 67 | - |
| *cythere* | - | 48 | - |
| *cytherelloidea* | *cytherelloidea cingulata* | 2 | - |
| *cytherelloidea* | - | 2 | - |
| *cytherelloidea* | *cytherelloidea yingliensis* | 3 | - |
| *cytherois* | *cytherois leizhouensis* | 199 | - |
| *cytherois* | - | 553 | - |
| *cytheropteron* | *cytheropteron cf. ignobilis* | 122 | - |
| *cytheropteron* | *cytheropteron higashikawai* | 1 | - |
| *cytheropteron* | *cytheropteron miurense* | 349 | - |
| *cytheropteron* | - | 114 | - |
| *cytherura* | - | 5 | - |
| *darwinula* | - | 2 | - |
| *eucythere* | - | 4 | - |
| *hanaiborchella* | *hanaiborchella cf. miurensis* | 77 | - |
| *hanaiborchella* | - | 7 | - |
| *haplocythereidea* | *haplocythereidea agilis* | 5 | According to the photos, should be *neocyprideis* |
| *haplocythereidea* | *haplocythereidea cf. agilis* | 2 | According to the photos, should be *neocyprideis* |

| Genus | Species | Count | Known problems |
|---|---|---|---|
| *hemicytheridea* | *hemicytheridea cancellata* | 12 | According to the photos, majority are *bicornucythere bisanensis* |
| *hemicytheridea* | *hemicytheridea reticulata* | 387 | - |
| *hemicytheridea* | - | 41 | - |
| *hemicytherura* | *hemicytherura cf. cuneata* | 121 | Possibile merge with *hemicytherura cuneata* |
| *hemicytherura* | *hemicytherura cf. kajiyamai* | 10 | Possibile merge with *hemicytherura kajiyamai* |
| *hemicytherura* | *hemicytherura cuneata* | 57 | Possibile merge with *hemicytherura cf. cuneata* |
| *hemicytherura* | *hemicytherura kajiyamai* | 14 | Possibile merge with *hemicytherura cf. kajiyamai* |
| *hemicytherura* | - | 131 | - |
| *hemikrithe* | *hemikrithe orientalis* | 158 | - |
| *hemikrithe* | *hemikrithe reticulata* | 2 | Typo of *hemicytheridea reticulata* |
| *hemikrithe* | - | 15 | - |
| *hermanites* | *hermanites bicostata* | 2 | - |
| *javanella* | *javanella kendengensis* | 8 | - |
| *javanella* | - | 1 | - |
| *keijella* | *keijella apta* | 1 | Possibile merge with *keijella cf. apta* |
| *keijella* | *keijella cf. apta* | 10 | Possibile merge with *keijella apta* |
| *keijella* | *keijella demissa* | 108 | Typo of *keijia demissa* |
| *keijella* | *keijella kloempritensis* | 2915 | - |
| *keijella* | - | 12 | - |
| *keijia* | *keijia demissa* | 9 | - |
| *kotoracythere* | *kotoracythere doratus* | 2 | - |
| *krithe* | *krithe japonica* | 28 | - |
| *krithe* | - | 1 | - |
| *leptocythere* | *leptocythere pulchra* | 6 | - |
| *leptocythere* | - | 3 | - |
| *loxoconcha* | *loxoconcha aff. hattorii* | 2 | Possibile merge with *loxoconcha hattorii* |
| *loxoconcha* | *loxoconcha aff. uranouchiensis* | 2 | Possibile duplicate with *loxoconcha cf. uranouchiensis* |
| *loxoconcha* | *loxoconcha aff. viva* | 7 | Possibile duplicate with *loxoconcha cf. viva* |
| *loxoconcha* | *loxoconcha cf. kattoi* | 82 | Possibile merge with *loxoconcha kattoi* |
| *loxoconcha* | *loxoconcha cf. kosugi* | 5 | - |
| *loxoconcha* | *loxoconcha cf. uranouchiensis* | 6 | Possibile duplicate with *loxoconcha aff. uranouchiensis* |
| *loxoconcha* | *loxoconcha cf. viva* | 1 | Possibile duplicate with *loxoconcha aff. viva* |
| *loxoconcha* | *loxoconcha epeterseni* | 149 | - |
| *loxoconcha* | *loxoconcha hattorii* | 11 | Possibile merge with *loxoconcha cf. hattorii* |
| *loxoconcha* | *loxoconcha japonica* | 221 | - |
| *loxoconcha* | *loxoconcha kattoi* | 145 | Possibile merge with *loxoconcha cf. kattoi* |
| *loxoconcha* | *loxoconcha malayensis* | 1177 | - |
| *loxoconcha* | *loxoconcha ocellata* | 1 | - |
| *loxoconcha* | *loxoconcha pulchra* | 16 | - |
| *loxoconcha* | - | 622 | - |

| Genus | Species | Count | Known problems |
|---|---|---|---|
| loxoconcha | loxoconcha zhejiangensis | 231 | - |
| macrocypris | - | 1 | - |
| microcythere | - | 8 | - |
| morkhovenia | morkhovenia inconspicua | 8 | - |
| munseyella | munseyella japonica | 341 | - |
| munseyella | munseyella oblonga | 3 | - |
| munseyella | - | 304 | - |
| neocyprideis | - | 13 | - |
| neocytheretta | neocytheretta elongata | 2 | Typo of *neosinocythere elongata* |
| neocytheretta | neocytheretta faceta | 160 | - |
| neocytheretta | neocytheretta snellii | 32 | - |
| neocytheretta | - | 408 | - |
| neocytheromorpha | neocytheromorpha regalis | 26 | - |
| neocytheromorpha | - | 34 | - |
| neomonoceratina | neomonoceratina delicata | 9650 | - |
| neomonoceratina | neomonoceratina elongata | 1 | Typo of *neosinocythere elongata*. However, according to photo, should be *spinileberis*. |
| neonesidea | neonesidea elegans | 210 | - |
| neonesidea | neonesidea oligodentata | 129 | - |
| neonesidea | - | 156 | - |
| neopellucistoma | neopellucistoma inflatum | 17 | - |
| neopellucistoma | - | 7 | - |
| neosinocythere | neosinocythere elongata | 1666 | - |
| neosinocythere | - | 799 | - |
| nipponocythere | nipponocythere bicarinata | 256 | - |
| nipponocythere | nipponocythere delicata | 632 | - |
| nipponocythere | - | 215 | - |
| orionina | orionina yongleensis | 1 | Species may be wrong |
| pacambocythere | - | 1 | - |
| palmenella | - | 8 | - |
| paracathaycythere | paracathaycythere cf. costaereticulata | 1 | - |
| paracypris | - | 71 | - |
| paracytheridea | paracytheridea tschoppi | 3 | - |
| paracytherois | paracytherois cf. acuminata | 45 | - |
| paracytherois | paracytherois cf. tosaensis | 2 | - |
| paracytherois | - | 51 | - |
| paradoxostomatid | - | 1257 | - |
| parakrithe | parakrithe cf. elongata | 1 | Could be *parakrithella* |
| parakrithe | parakrithe japonica | 1 | - |
| parakrithella | parakrithella cf. pseudadonta | 26 | Possibile merge with *parakrithella pseudadonta* |
| parakrithella | parakrithella pseudadonta | 2 | Possibile merge with *parakrithella cf. pseudadonta* |
| parakrithella | - | 8 | - |
| phlyctocythere | phlyctocythere japonica | 328 | - |
| phlyctocythere | - | 142 | - |
| pistocythereis | pistocythereis aff. miaoliensis | 12 | - |
| pistocythereis | pistocythereis bradyformis | 1015 | - |
| pistocythereis | pistocythereis bradyi | 1758 | Possibile merge with *pistocythereis cf. bradyi* |
| pistocythereis | pistocythereis cf. bradyi | 41 | Possibile merge with *pistocythereis bradyi* |

| Genus | Species | Count | Known problems |
|---|---|---|---|
| *pistocythereis* | *pistocythereis cf. subovata* | 11 | Possibile merge with *pistocythereis subovata* |
| *pistocythereis* | *pistocythereis euplectella* | 74 | - |
| *pistocythereis* | *pistocythereis subovata* | 56 | Possibile merge with *pistocythereis cf. subovata*, very hard to be distinguished from *pistocythereis bradyi* visually. |
| *pistocythereis* | - | 1884 | - |
| *pontocythere* | *pontocythere cf. subjaponica* | 44 | - |
| *pontocythere* | - | 96 | - |
| *propontocypris* | *propontocypris clara* | 1 | - |
| *propontocypris* | - | 5021 | - |
| *pseudocythere* | *pseudocythere cf. caudata* | 1 | - |
| *pseudocythere* | - | 5 | - |
| *robustauria* | *robustauria cf. ishizakii* | 34 | - |
| *robustauria* | *robustauria salebrosa* | 2 | - |
| *robustauria* | - | 5 | - |
| *semicytherura* | *semicytherura cf. miurensis* | 12 | - |
| *semicytherura* | *semicytherura cf. undata* | 1 | - |
| *semicytherura* | *semicytherura cf. wakamurasaki* | 3 | - |
| *semicytherura* | *semicytherura indonesiana* | 6 | - |
| *semicytherura* | - | 120 | - |
| *sinocythere* | *sinocythere dongtaiensis* | 3 | - |
| *sinocythere* | *sinocythere sinensis* | 87 | - |
| *sinocythere* | - | 183 | - |
| *sinocytheridea* | *sinocytheridea impressa* | 22429 | - |
| *sinoleberis* | *sinoleberis cf. tosaensis* | 3 | - |
| *spinileberis* | *spinileberis quadriaculeata* | 1484 | - |
| *spinileberis* | *spinileberis quadriculeata* | 1 | Typo of *spinileberis quadriaculeata* |
| *spinileberis* | *spinileberis rhomboidalis* | 342 | - |
| *spinileberis* | - | 111 | - |
| *stigmatocythere* | *stigmatocythere aff. roesmani* | 8 | Possibile duplicate with *stigmatocythere cf. roesmani* |
| *stigmatocythere* | *stigmatocythere bona* | 78 | - |
| *stigmatocythere* | *stigmatocythere cf. roesmani* | 17 | Possibile duplicate with *stigmatocythere aff. roesmani* |
| *stigmatocythere* | *stigmatocythere costa* | 58 | - |
| *stigmatocythere* | *stigmatocythere kingmai* | 64 | - |
| *stigmatocythere* | *stigmatocythere roesmani* | 767 | Possibile merge with *stigmatocythere cf. roesmani* and *stigmatocythere aff. roesmani* |
| *stigmatocythere* | - | 270 | - |
| *tanella* | *tanella gracillis* | 81 | - |
| *tanella* | - | 29 | - |
| *trachyleberididae* | - | 25 | - |
| *trachyleberis* | - | 2 | - |
| *triebelina* | *triebelina aff. sertata* | 8 | - |
| *xestoleberis* | *xestoleberis hanaii* | 222 | - |
| *xestoleberis* | *xestoleberis suetsumuhana* | 2 | - |
| *xestoleberis* | - | 860 | - |
| *xiphichilus* | - | 87 | - |
| | *Grand Total* | 71466 | |

### 3.3 Version Difference

The images in the *Noisy Ostracods* dataset and its 2022 version do not have a one-to-one correspondence. This means that images with the same name may not be the same across versions. The reason is straightforward: we performed per-image re-labeling when building the *Noisy Ostracods* dataset based on the 2022 version to identify and correct possible false labels and missing labels.

We provide the original train, test, and validation splits from 2022. However, this version has the following issues:

- **Random Split**: The splits for genus and species are not consistent. This means that the test and validation images at the genus and species levels do not match. Some images in the test set for genus identification may appear in the training set for species identification.

- **Minor Class Elimination**: All minor classes with fewer than 10 images were removed rather than being moved to the training set, unlike in the current version.

Please consider these differences when comparing methods using the 2022 version of the dataset.

## 4 Data Availability and Maintenance

We are making the Noisy Ostracods datasets and the Noisy Ostracods 2022 datasets available online.

Noisy Ostracods images: Noisy Ostracods images: Click to download

Noisy Ostracods 2022 images: Click to download

Croissant[9] metadata: Click to download

Code: `https://github.com/H-Jamieu/Noisy_ostracods`

We are actively following the taxonomy changes in ostracods and will revise the taxonomy accordingly. New samples from our studies will be annotated and published after verification. Any future updates on the dataset will be included in the code repository. We are currently cleaning the entire dataset and discussing potential changes in some ambiguous species. The link to download the fully cleaned dataset will be released on the code repository. One copy of the dataset will be hosted in the Data Repository of The University of Hong Kong to ensure long term preservation.

## 5 Dataset Licence and Author Responsibility

The authors hereby declare that they bear all responsibility in case of violation of rights, including but not limited to intellectual property rights, data privacy rights, and any other applicable laws. They confirm that the data provided in this work is licensed under the Creative Commons Attribution 4.0 International (CC BY 4.0) license[10].

## 6 Additional Contents

### 6.1 Error Correction

During the manual cleaning of the dataset, we initially reported the discovery of two new genera, including *Psudocythere* and another unnamed genus. Upon further inspection, we realized that *Psudocythere* was already present in our dataset. This oversight occurred due to an error in our initial data review process. We deeply apologize for this mistake and any confusion it may have caused.

Furthermore, we are not certain if the specimen is indeed *Psudocythere* since the key identification parts of the specimen are broken. We are still checking the relevant taxonomic materials to address this issue at the time of writing. Meanwhile, the other specimen is confirmed to be a new genus.

As a result, the images are still being deleted and will not affect the reported results. The two images in question are `surface_VS13_6_ind4.tif` and `rawSample_B1a_29_ind2.tif`, for users of the dataset. The corresponding paragraph has been revised in our latest version of the main article.

## 6.2 Additional Hyper-parameters

In the main article, we forgot to mention the size of the images for training. We used 224*224 for all the training. For Co-teaching[11], Co-teaching+[12], loss-clip, mixup-cutmix[13, 14], CL[15] and CE training, we used FP16 scaling to accelerate training. As for Divide-mix[16], we follow the official implementation not scaling the training to FP16. For embedding calculated using CLIP-ViT-L-14@336[17], we scaled the images to 336*336 to perserve the consistency. All other embeddings are calculated using the image size the models trained on for SimiFeat[18].