# OpenReview forum: "Noisy Ostracods: A Fine-Grained, Imbalanced Real-World Dataset for Benchmarking Robust Machine Learning and Label Correction Methods"
_NeurIPS.cc/2024/Datasets_and_Benchmarks_Track — NeurIPS 2024 Track Datasets and Benchmarks Poster_

### Official Review · Reviewer_mtcB · 2024-07-09
**Noisy Ostracods: Real-World Dataset for Benchmarking Robust Machine Learning and Label Correction Methods**

**Rating:** 7
**Confidence:** 4
**Correctness:** Yes
**Clarity:** Yes

**Review:**

Real-world datasets always present some interesting aspects when compared to academia ones. Ostracod dataset has a number of "hard" challenges that can be used to benchmark various ML methods. Therefore, motivation of this dataset is clear and appealing. Authors did a good job constructing dataset and splits with the assumptions they had. Also, authors evaluated a number of popular baselines for robust learning which can be used later by new methods. In summary, this is a solid work.

**Strengths:**

1. Real-world images with relatively "hard" classification task.
2. Assumptions about data including label noise, imbalance and other limitations are accomplished.
3. Popular baselines for robust learning including label error detections methods have been evaluated.
4. The data collection pipeline is described in details.

**Additional Feedback:**

Please respond to my comments

**Documentation:**

Yes (the GitHub link is not public yet, but everything is provided in the supplementary materials)

**Limitations:**

Yes

**Opportunities For Improvement:**

1. Few minor typos e.g., @L196 "o ensure".
2. The provided link to the dataset page is not working. When do you plan to make it public?
3. Could you think about other types of applications? For example, unsupervised anomaly detection etc.?
4. Do you plan to have a cleaned data by the conference time?

**Relation To Prior Work:**

Yes

**Summary And Contributions:**

This dataset contains a collection of ostracod images with noisy labels. Classification of ostracod species is a quite challenging task. This is especially true when labels are noisy, taxonomy is evolving and classes are imbalanced. In addition, authors evaluated several popular baselines to deal with the label noise. Overall, the proposed dataset seems to be a good contribution to the research community.

---

> ### Author Rebuttal · Authors · 2024-08-17
>
> # Respond to rviewer's comments.
> We sincerely thank you for your thorough and insightful review of our paper "Noisy Ostracods: Real-World Dataset for Benchmarking Robust Machine Learning and Label Correction Methods". We appreciate your recognition of our dataset's value and contribution to the research community. We are grateful for your positive feedback on the dataset's real-world challenges, our detailed data collection pipeline, and the evaluation of popular baselines. Your comments affirm our goal of providing a meaningful benchmark for robust learning methods. And I am going to answer your questions:
>
> # 1. Typo
> We are currently in our second revision post-submission, with two more revisions planned this month. We are committed to significantly reducing typos in the final version.
>
> # 2. Link to download not working and data publicy plan
>
> We apologize for the insufficient testing of documentation links. We've identified that links work only within our university network. However, links within the metadata file should be functional. We will post working links in the global comment section for all authors. Regarding public availability, the dataset will be accessible no later than 1, December 2024.
>
> # 3. Other application of the dataset
>
> We appreciate your suggestion and have identified several potential applications:
> a) Unsupervised Anomaly Detection: This could be used to filter out low-quality instances (feature errors) before applying Label Noise Learning methods for label error correction.
>
> b) Long-Tailed Learning: As noted by reviewer Q9DE, our extremely imbalanced dataset is suitable for this research area.
>
> c) Distribution Shift: We're collecting ~100,000 new images using 100X lenses, which produce a 'purple' shift compared to our current 40-50X lens images. This presents opportunities for research on model robustness to distribution shifts.
>
> d) Novel Category Discovery [1]: Exploring how this could aid in identifying mislabeled new species.
>
> e) Out-of-Distribution Detection: Investigating its potential for new species identification.
>
> f) Few-shot/One-shot Learning: Our dataset's rare species provide a testbed for these methods.
>
> # 4. Clean data by conference time
>
> We aim to release our progress on a cleaned version of the dataset by 31, December 2024.
>
> We appreciate your constructive feedback and will incorporate these improvements in our final submission.
>
> [1]. Automatically discovering and learning new visual categories with ranking statistics. K Han, SA Rebuffi, S Ehrhardt, A Vedaldi, A Zisserman

---

> > ### Comment · Reviewer_mtcB · 2024-08-19
> > **rebuttal**
> >
> > Thank you for your rebuttal. I'd like to keep my Accept rating.

---

> ### Author Response · Authors · 2024-08-20
> **Rseopse to the reviewer**
>
> We are delighted to hear you continued the Accept rating!
>
> We're actively exploring new approaches to enhance our dataset and methods:
>
> 1. Unsupervised Anomaly Detection:
> * Implementing DBSCAN as our initial method.
> * Investigating Image Quality Assessment techniques for feature error detection.
> We look forward to sharing these results with you soon.
> 2. Integrating Unsupervised and Self-Supervised Methods in Label Noise Learning:
> Our proposed approach:
> a) Pretrain unsupervised models.
> b) Identify exemplar instances (anchor point) for each class with expert assistance.
> c) Perform clustering using features learned by unsupervised models.
>
> Potential benefits:
> * Leverages the entire dataset, including noisy instances, for feature learning.
> * Significantly reduces the effort required compared to cleaning the entire dataset.
> * Outcome depends on the representation power of the feature extractor and expert selection of exemplars.
>
> Current challenges:
> * Unexpectedly poor clustering performance from CLIP[1] and MAE[2] as reported in the paper. This is the main cause of poor performance of SimiFeat.
>
> We are woriking on:
> * Exploring DINO[3] and MOCO[4] series as alternatives.
> * Investigating the hypothesis that supervised fine-tuning may be crucial due to our dataset's small size and the rarity of ostracods in pre-training datasets.
>
> We're conducting experiments to validate these hypotheses and will update you on our findings.
>
> **References**
>
> [1] Alec Radford et al. Learning Transferable Visual Models From Natural Language Supervision.
> 2021. arXiv: 2103.00020 [cs.CV].
> [2] Kaiming He et al. “Masked Autoencoders Are Scalable Vision Learners”. In: arXiv:2111.06377 (2021).
> [3] Oquab et al., (2024). DINOv2: Learning Robust Visual Features without Supervision. arXiv preprint arXiv:2304.07193.
> [4] He et al. (2019). Momentum Contrast for Unsupervised Visual Representation Learning. arXiv preprint arXiv:1911.05722.

---

### Official Review · Reviewer_Q9DE · 2024-07-20
**An interesting dataset but may not be that useful for the area of label-noise learning**

**Rating:** 6
**Confidence:** 4
**Correctness:** This dataset is technically sound.
**Clarity:** Yes. The paper is well-organized and …

**Review:**

Pros:
1.	This dataset makes up for the insufficiency of real-world datasets with noisy labels.
2.	A dataset with both feature error and annotation error is new to the community. It will attract researchers for a potentially new category of methods.
Cons:
1.	If this dataset is imbalanced, it will also be interesting to the community of imbalance/long-tail learning. The authors should broaden the group of researchers who will potentially use this dataset.
2.	This dataset may not be suitable for being a benchmark for general image classification algorithms because a model with good performance on the task of ostracod classification may not imply that it also performs well on other image classification tasks.
3.	The methods listed and compared in Table 1 are dated. The latest one Divide-mix, although it is one of the typical methods in label-noise learning, was proposed in 5 years ago.
4.	Proposing a new method for the new benchmark is good but not necessary for a benchmark paper. There can be more space to discuss the property of the new dataset.

**Strengths:**

See Pros in Review.

**Additional Feedback:**

N/A

**Documentation:**

The authors provide detailed documentation and Github.

**Limitations:**

1.	This work has no potential negative societal impact.
2.	The diversity of a dataset about genus and species classification of crustacean ostracods is low. There will not many potential researches that will use this dataset as benchmark to evaluate the model performance. I suggest the authors directly propose new label-noise learning methods for this dataset, rather than making it a benchmark.

**Opportunities For Improvement:**

Please continue to update and maintain the GitHub repository. Additionally, it’s highly recommended to consider enabling permissions to allow other users to contribute to the repository, fostering a collaborative and dynamic development environment.

**Relation To Prior Work:**

Yes. This work covers appropriate related works, but not too much.

**Summary And Contributions:**

This paper introduces a new Noisy Ostracods dataset, designed for genus and species classification of crustacean ostracods with annotations from specialists. It includes open-set noise and pseudo-classes, presenting a unique challenge for robust machine learning methods. This dataset will draw certain attention from researchers in the area of label-noise learning.

---

> ### Author Rebuttal · Authors · 2024-08-17
>
> ## (Reply to the reviewer 1/3)
> Thank you for your thorough review of our paper introducing the Noisy Ostracods dataset. We appreciate your insights and would like to address your comments while clarifying some potential misunderstandings about the purpose and scope of our work.
>
> # Background Information
> Accurate taxonomy is crucial in biological research, but integrating machine learning (ML) into workflows often faces challenges due to researchers' unfamiliarity and mistrust of these systems. A primary cause of this mistrust is the presence of errors in training data. Even with 95% accuracy, the remaining 5% of mislabeled data can significantly undermine confidence in model predictions. Detecting and correcting these final errors is exceptionally challenging, as specialists often make consistent mistakes due to overfamiliarity, and external experts are reluctant to review large datasets manually.
>
> This situation underscores the critical need for effective noisy label learning methods. However, our experiments with the Noisy Ostracods dataset revealed that:
>
> * Current robust ML methods didn't significantly outperform vanilla CE training on noisy data.
> * Existing label correction methods were insufficient for addressing label errors in the dataset.
>
> These challenges may stem from the fine-grained, long-tail nature of the dataset. By publishing this dataset, we aim to draw attention to the urgent need for improved label correction methods in practical biological research, potentially bridging the gap between ML capabilities and researchers' trust in these systems.
>
> # Pros of the dataset
> > Pros: 1. This dataset makes up for the insufficiency of real-world datasets with noisy labels. 2. A dataset with both feature error and annotation error is new to the community. It will attract researchers for a potentially new category of methods.
>
> We appreciate your recognition of the Noisy Ostracods dataset's contributions to the field. This dataset addresses a critical gap in the Learning with Noisy Labels (LNL) research by providing a real-world alternative to the commonly used variants of CIFAR-10 and ImageNet. Its unique characteristics, particularly the coexistence of feature errors and label errors, present novel challenges for state-of-the-art LNL methods.
>
> The dataset's distinctive attributes include:
>
> 1. Authentic label noise derived from real-world taxonomic challenges.
> 2. Comprehensive analysis of noise sources, facilitating more realistic assumptions in LNL applications.
> 3. A combination of feature and label errors, reflecting the complexities of real-world data.
> 4. Co-existing of pesudo class and un-discovered new classes.
>
> These features collectively offer a more diverse and challenging evaluation framework for LNL methods, complementing existing benchmarks. By providing this dataset, we aim to stimulate the development of new categories of methods that can effectively address the multifaceted nature of label noise in practical applications.
>
> Furthermore, the detailed documentation of noise sources and types enables researchers to formulate more realistic hypotheses and develop methods with greater applicability to real-world scenarios. This aligns with your observation that the dataset has the potential to attract researchers towards developing novel categories of LNL methods.
>
> # Adressing cons
> >  1. If this dataset is imbalanced, it will also be interesting to the community of imbalance/long-tail learning. The authors should broaden the group of researchers who will potentially use this dataset.
>
> We appreciate this insightful observation. Indeed, we are aware of the dataset's imbalanced nature, which reflects real-world biodiversity patterns. We included 'imbalanced' in the article's title to attract relevant researchers.
>
> Notably, we identified one study addressing imbalanced label noise problems [1]. However, it used synthetic unbalanced data, highlighting the lack of real imbalanced noisy datasets in academia. We plan to implement this method to demonstrate our dataset's suitability for addressing imbalanced label noise problems in real-world scenarios.
>
> >  2. This dataset may not be suitable for being a benchmark for general image classification algorithms because a model with good performance on the task of ostracod classification may not imply that it also performs well on other image classification tasks.
>
> We appreciate this observation and agree that the Noisy Ostracods dataset is not intended as a benchmark for general image classification algorithms. Our primary objective is to provide a challenging, real-world testbed for Learning with Noisy Labels (LNL) methods, rather than for broad image classification tasks. The noise types in our dataset are highly diverse, reflecting real-world challenges in taxonomic classification. Importantly, we've observed that methods which performed well on previous noisy datasets, such as CIFAR-10, did not perform equally well on our dataset. This discrepancy highlights a critical generalization issue in the field of LNL research.
>
> Our findings suggest that:
>
> 1. Previous benchmark datasets may not capture the full complexity of real-world label noise.
> 2. There's a need for more noise-diverse, domain-specific datasets to test the robustness and generalizability of LNL methods.
>
> By providing a dataset with complex, real-world noise patterns, we aim to:
>
> * Expose limitations in current LNL methods when applied to challenging, domain-specific tasks.
> * Bridge the gap between theoretical LNL research and practical applications in fields like biological taxonomy.
>
> We believe our dataset offers unique value in advancing LNL research towards more realistic and challenging scenarios.

---

> > ### Comment · Reviewer_Q9DE · 2024-08-22
> >
> > Thanks for the authors' rebuttal. My concerns are partially solved. I'm willing to increase my score to 6.

---

> > > ### Author Response · Authors · 2024-08-22
> > > **Response to the reviewer**
> > >
> > > We are delighted to hear part of the concerns are adressed. We will keep updating the dataset hoping other concerns can be addressed as well!
> > >
> > > Thank you very much!

---

> ### Author Rebuttal · Authors · 2024-08-17
>
> ## (Reply to the reviewer 2/3)
>
> > 3. The methods listed and compared in Table 1 are dated. The latest one Divide-mix, although it is one of the typical methods in label-noise learning, was proposed in 5 years ago.
>
> Addressing the benchmarking concerns, we have significantly expanded our evaluation by implementing four recent algorithms proposed after 2021:
>
> 1. Sharpness-Aware Minimization (SAM) [2] (2021)
> 2. Part-Level Multilabeling (PLM) [3] (2024)
> 3. Subclass-Dominant Label Noise (SDN) [4] (2023)
> 4. Label Wave [5] (2024)
>
> Results are listed in Table 1. For justifications and implementation details please refer to the reply we given to Reviewer x9Zo.
> | Method          | Acc          | Prec.         | Recall        | F1 Score      |
> |-----------------|--------------|---------------|---------------|--------------|
> | CE              | 95.77 ± 0.35 | 83.31 ± 4.02  | 75.20 ± 2.78  | 76.51 ± 2.99 |
> | SAM \[2\]       | 94.49 ± 0.29 | 77.34 ± 1.69  | 69.70 ± 2.16  | 70.55 ± 1.89 |
> | PLM \[3\]       | 96.70 ± 0.06 | 79.22 ± 1.75  | 72.81 ± 1.81  | 74.09 ± 1.65 |
> | SDN \[4\]       | 93.91 ± 1.13 | 75.92 ± 5.02  | 69.33 ± 2.55  | 69.80 ± 3.60 |
> | CE-label Wave \[5\] | 95.21 ± 0.23 | 84.23 ± 2.78  | 73.66 ± 2.15  | 75.29 ± 2.34 |
>
> Table 1. The accurancy, precision, recall and F1 Score of each methods with ResNet-50 backbone. For details  please refer to the reply we given to Reviewer x9Zo.
>
> These results demonstrate that our dataset presents significant challenges even for recent state-of-the-art methods, highlighting its value as a benchmark for advancing Learning with Noisy Labels research.
>
> > 4. Proposing a new method for the new benchmark is good but not necessary for a benchmark paper. There can be more space to discuss the property of the new dataset.
>
> We appreciate the reviewer's perspective. While proposing a new method isn't essential for a benchmark paper, we believe our work balances dataset analysis with methodological contribution.
>
> We developed Naive Ensemble Cross Validation (NECV) as an original approach to address our dataset's unique challenges, only later discovering similar methods in literature. Regarding dataset properties, we've provided comprehensive analyses of label noise types, their sources, and feature clusterability. We believe this level of detail, combined with NECV, offers crucial insights for researchers and aligns with the primary objective of a dataset paper. We're open to suggestions for expanding specific aspects of the dataset property discussion if needed.
>
> # Github open access
> We followed the reviewer's advice and made the github open access. We are committed to refactoring the code into a more user-friendly format. Since the supplementary material version, we've added new methods to the repository, including Sharpness Aware Minimization [2] and Label Wave [5], as discussed in our results section. Following reviewer x9Zo's advice, implementations of Subclass Dominate Error [4] and PLM [5] are currently in separate repositories to maintain organization. These will be integrated during the code refactoring. We enthusiastically welcome contributions from the community.
>
> # Adressing Limitations
> > 1. This work has no potential negative societal impact.
>
> We noticed that the absence of negative social impact was listed as a limitation. We consider this a positive attribute, reflecting our commitment to ethical research practices. This aligns with our goal to provide a valuable resource to the community without any adverse effects.
>
> > 2. The diversity of a dataset about genus and species classification of crustacean ostracods is low. There will not many potential researches that will use this dataset as benchmark to evaluate the model performance. I suggest the authors directly propose new label-noise learning methods for this dataset, rather than making it a benchmark.
>
> We appreciate the reviewer's thoughtful comments on our dataset. We'd like to clarify that while the dataset's focus on genus and species classification of crustacean ostracods may seem narrow from a general image classification perspective, this specificity is actually unrelated to its primary purpose and value in Learning with Noisy Labels (LNL) research.
>
> Our dataset was designed specifically to address challenges in LNL, not as a general-purpose image classification benchmark. In the context of LNL research, the key properties of our dataset are:
>
> 1. Real-world Noise Patterns: The dataset captures authentic label noise that occurs in expert-annotated biological data, providing a realistic testbed for LNL methods.
> 2. Diverse Noise Types: Despite its domain specificity, the dataset exhibits a rich variety of noise patterns relevant to LNL research across various fields.
> 3. Fine-grained Classification Challenges: The complexity of distinguishing between closely related species presents unique challenges for LNL methods, pushing the boundaries of current techniques.
> 4. Practical Relevance: The dataset addresses a real need in biological data cleaning, demonstrating the practical applications of LNL research in scientific domains.
>
> These properties make our dataset particularly valuable for advancing LNL research, regardless of its scope in terms of general image classification. The dataset's strength lies in its ability to test and inspire LNL methods that can handle complex, real-world noise in specialized domains - a critical area for the field's progression.
>
> Given the unique properties of our dataset - real-world noise patterns, diverse noise types, fine-grained classification challenges, and practical relevance - we believe it serves as a valuable benchmark for LNL research. We also proposed our baseline, NECV for future researchers to compare. By making it available, we aim to facilitate and inspire innovative solutions from the broader research community.

---

> ### Author Rebuttal · Authors · 2024-08-17
>
> ## (Reply to the reviewer 3/3)
>
> # Adressing misunderstandings about the dataset
>
> We appreciate the thoughtful review and acknowledge the valuable feedback provided. We'd like to address the key points raised and clarify any potential misunderstandings about the usefulness:
>
> 1. We're grateful for the reviewer's recognition of our dataset's potential contributions to the Learning with Noisy Labels (LNL) research community, particularly highlighting its filling the gap of lack of real noisy dataset in the field.
> 2. We've addressed the concerns regarding open-source availability on GitHub and the implementation of state-of-the-art methods, enhancing the dataset's accessibility and relevance to current research trends.
> 3. While we appreciate the additional concerns raised about the dataset's suitiablity as a general image classification benchmark, we note that they primarily relate to aspects outside the core focus of LNL research. Our dataset was specifically designed to address challenges in LNL, and its strengths lie in this domain.
>
> Given the about points, we believe there may be some misunderstanding regarding the target and primary purpose of our dataset. Our dataset is specifically designed for and highly relevant to the area of LNL. The acknowledged strengths in addressing real-world noise patterns and providing novel challenges for LNL methods directly align with our intended goals and underscore its significant value as a benchmark in this field. The unique characteristics of our dataset, particularly its diverse and authentic noise types, offer substantial benefits to the LNL research community by presenting challenges that are more representative of real-world scenarios than many existing benchmarks.
>
> We remain committed to further improving and expanding the dataset's utility for LNL research and are open to additional suggestions that align with this core focus.
>
> ## References
> [1] S. Jiang, J. Li, J. Zhang, Y. Wang and T. Xu, "Dynamic Loss for Robust Learning," in IEEE Transactions on Pattern Analysis and Machine Intelligence, vol. 45, no. 12, pp. 14420-14434, Dec. 2023, doi: 10.1109/TPAMI.2023.3311636
> [2] Foret, P., Kleiner, A., Mobahi, H., & Neyshabur, B. (2021). Sharpness-aware Minimization for Efficiently Improving Generalization. In International Conference on Learning Representations.
> [3] Zhao, R., Shi, B., Ruan, J., Pan, T., & Dong, B. (2024). Estimating Noisy Class Posterior with Part-level Labels for Noisy Label Learning. In Conference on Computer Vision and Pattern Recognition.
> [4] Bai, Y., Han, Z., Yang, E., Yu, J., Han, B., Wang, D., & Liu, T. (2023). Subclass-Dominant Label Noise: A Counterexample for the Success of Early Stopping. In Thirty-seventh Conference on Neural Information Processing Systems.
> [5] Yuan, S., Feng, L., & Liu, T. (2024). Early Stopping Against Label Noise Without Validation Data. In The Twelfth International Conference on Learning Representations.

---

### Official Review · Reviewer_GKNX · 2024-07-25
**An intriguing dataset paper for studying robustness to label noise**

**Rating:** 7
**Confidence:** 4

**Review:**

This is a well written and motivated paper. I am happy to accept. The only obvious issue I see is that the github link it broken; however, the code for review was provided in the supplemental and links to the dataset were also found within the supplemental.

Pros:
- Well written
- Well motivated
- Noise analysis is thorough and informative
- The dataset is adequately described (the supplemental aids this)
- Experiments are easy to understand and demonstrate the difficulty of using LNL methods / the lack of effectiveness of these methods in real world usage, as well as the importance of furthering the research on handling data with label errors.

Cons:
- Did not mention other similar ostracod datasets such as the World Ostracoda Database (https://www.marinespecies.org/ostracoda/) and iioe_ostracods (https://www.bco-dmo.org/dataset-deployment/451987) - although the purpose of this dataset paper is to provide a test bed for handling noise, rather than specifically to study ostracods.
- Link to github in paper did not work.

**Strengths:**

Similar to 'pros' in main review.

This paper represents a meaningful contribution in that it is both realistic and that LNL methods are not able to solve the problem of noisy labels. By bringing this issue to light, in a simple but clear way, one hopes that more researchers from the broader ML community will become aware and interested in the importance of handling noisy data.

**Additional Feedback:**

None.

**Clarity:**

The paper is well written and components such as figures, tables, and the algorithm also also presented clearly.

**Correctness:**

The claims made appear to be correct. The dataset seems to be constructed in a sound way. Benchmark evaluation methods appear to be appropriate.

**Documentation:**

Sufficient details is given.

**Ethics:**

No concerns.

**Limitations:**

Limitations were adequately discussed.

**Opportunities For Improvement:**

See 'cons' of main review. Be sure to fix link to github.

It would be nice to see some of this analysis extended to other datasets. In some ways, this work is not really about the ostracod dataset, but rather the issue of contending with label errors. This dataset provides a testbed for experimenting on this task, but perhaps additional framing and experiments could help.

**Relation To Prior Work:**

The differences to prior work (other datasets) and even the work done by other LNL papers is discussed.

**Summary And Contributions:**

This paper presents a realistic dataset from a realistic problem whereby the seemingly inevitable presence and susceptibility to label errors causes difficulties. In particular, a dataset of ostracods was collected over a period of 10 years with the purpose of studying the anthropogenic impact on Hong Kong marine environments. However, label errors have plagued this dataset, in spite of continued efforts to clean it. This prompted questions on the utility of using "Learning with Noisy Labels (LNL)" methods vs standard training methods when label errors are present, and measuring how many label errors can be detected by existing LNL methods vs a proposed cross-validation-based baseline. Related biological datasets with probable label noise are presented. The types of noise in this data are presented in detail. In particular, two broad categories are used: Feature error (which lead to indiscernibility) and Label error (explicitly an error derived from the time of labelling, but as a result of various causes, such as some classes just being difficult). Experiments are performed for robust learning methods and label correction methods, seeking to answer the two main proposed questions. A reasonable set of limitations are discussed, and then paper concludes with plans to take this project further.

---

> ### Author Rebuttal · Authors · 2024-08-17
>
> # Response to the Reviewer
> We sincerely thank you for your insightful review. Your summary of our work is remarkably concise and accurately captures all the key points we aimed to emphasize about the dataset. We greatly appreciate your recognition of our work's motivation, writing quality, and contributions to the field.
>
> ## Addressing the Cons
> We're grateful for your constructive feedback and would like to address the points you raised:
>
> ## Comparison with other ostracod datasets:
> Thank you for bringing the World Ostracoda Database and iioe_ostracods dataset to our attention. We'd like to clarify the distinctions:
> * The World Ostracods Database primarily contains a name list of ostracods with links to their taxonomic descriptions.
> * The iioe_ostracods dataset focuses on textual data about planktonic ostracods, which differs from the benthic ostracods covered in our dataset.
> * Neither of these datasets provides the extensive image collection of specific species that our dataset offers.
> We will include a brief discussion of these datasets in our revised manuscript to provide additional context.
> ## GitHub link:
> The repository is now public, and we are actively updating it. We will ensure the correct, functioning link is provided in the final version of the paper.
>
> ## Improvements
> As per reviewer x9Zo's request, we have added more experiments on state-of-the-art methods.
>
> Regarding the suggestion to extend our analyses to other datasets, we've identified an interesting opportunity. We've noticed that no one has attempted to fix CIFAR-10N[1] using Confident Learning[2]. We're interested in applying our metrics to test CIFAR-10 label errors using CL. We plan to pursue this during the discussion period and will keep you updated on our progress.
>
> We appreciate your positive evaluation and your helpful suggestions for improvement. We will incorporate these changes in our final submission to strengthen the paper's impact and clarity.
>
> Thank you again for your valuable input and positive recommendation.
>
> [1] Learning with noisy labels revisited: A study using real-world human annotations, 2022 ICLR
> [2] Curtis G. Northcutt, Lu Jiang, and Isaac L. Chuang. “Confident Learning: Estimating Uncer351 tainty in Dataset Labels”. In: Journal of Artificial Intelligence Research (JAIR)

---

> > ### Comment · Reviewer_GKNX · 2024-08-30
> >
> > Thank you for your detailed response addressing my concerns. I see that the link to the GitHub repo is now working. I feel that the improvements and revisions you have mentioned will strengthen your paper and satisfy my concerns.

---

> > > ### Author Response · Authors · 2024-08-31
> > > **Response to the reviewer**
> > >
> > > Thank you for your comment! We are very glad to hear the concerns are adressed.

---

### Official Review · Reviewer_x9Zo · 2024-07-25
**The collected dataset has a good contribution, but the benchmark experiments and writing need to be improved.**

**Rating:** 7
**Confidence:** 4
**Correctness:** For the benchmark, please check the O…
**Clarity:** Please check the Opportunities For Im…

**Review:**

Please check the comments below.

**Strengths:**

The collected dataset is valuable considering the sample collection over the last 10 years and domain-specific fine-grained categories.

**Additional Feedback:**

The reviewer will consider raising the score if the concerns can be addressed.

**Documentation:**

Yes

**Limitations:**

Yes

**Opportunities For Improvement:**

## Weaknesses
- Benchmarking.
1. The number of methods used for benchmarking noisy label learning is relatively few. Recent new datasets [1] [2] that focus on noisy label learning generally benchmark more than 10 methods.
[1] Learning with noisy labels revisited: A study using real-world human annotations, 2022 ICLR
[2] MVP-N: A Dataset and Benchmark for Real-World Multi-View Object Classification, 2022 NeurIPS
2. It would be better to include all the label correction methods provided in AQuA [3] rather than only using Confident Learning and SimiFeat.
[3] AQuA: A Benchmarking Tool for Label Quality Assessment, 2023 NeurIPS
3. Reproducibility and maintainability. Based on the code provided in the supplementary material, the code structure is not well-organized and needs to be implemented in a cleaner version. The current version is not good enough to be maintained and used by other researchers. Also, the authors do not provide the checkpoints for the reviewers to check the reproducibility. The GitHub link cannot be accessed, and there is no leaderboard.
- Proof reading is required since there are many typos.

**Relation To Prior Work:**

Yes

**Summary And Contributions:**

This study proposes a fine-grained noisy dataset for crustacean ostracod classification with specialists’ annotations for benchmarking noisy label learning and label correction methods.

---

> ### Author Rebuttal · Authors · 2024-08-16
>
> ## (Reply to reviewer 1/3)
> Thank you for your insightful review and for recognizing the value of our dataset. We appreciate your acknowledgment of the Noisy Ostracods dataset's contribution to the field. Our primary goal in introducing this dataset is to provide a challenging benchmark for Label Noise Learning (LNL) methods that encompasses diverse noise types in a real-world context. The dataset's unbalanced and fine-grained nature presents unique challenges to the LNL community, pushing the boundaries of current techniques. Furthermore, we aim to stimulate research on label correction methods, which have significant practical applications. As our experience demonstrates, cleaning the final 5% of noisy instances in the dataset is extremely challenging and time-consuming, even for domain specialists. We believe this dataset will drive innovations in robust machine learning and label correction techniques. We appreciate your feedback and will address your concerns in detail in this response.
>
> # 1. Expanded Benchmarks on Learning with Label Noise Methods
> Addressing the benchmarking concerns, we have significantly expanded our evaluation by implementing four recent algorithms proposed after 2021:
> 1. Sharpness-Aware Minimization (SAM) [1]
> 2. Part-Level Multilabeling (PLM) [2]
> 3. Subclass-Dominant Label Noise (SDN) [3]
> 4. Label Wave [4]
>
> Our selection of these methods was deliberate and aligned with the unique characteristics of our dataset:
>
> * SAM was chosen based on recent intriguing analyses [5] discussing its robustness to label noise.
> * PLM was selected for its potential in addressing label noise through part-level labels, which could be particularly beneficial for our fine-grained classification task.
> * SDN was included as it perfectly matches a specific noise type in our dataset: subclass-dominant noise. Ostracods have up to 9 juvenile stages, each differing from the adult stage, forming latent subclasses. For instance, juvenile stages of Alocopocythere and Neocytheretta are easily confused, while their adult stages are more distinguishable, exemplifying subclass-dominant error.
> * Label Wave was added for its mechanism of finding better early stopping points to avoid memorization of noisy instances.
>
> Notably, SDN suggests late stopping when encountering subclass-dominant errors, while Label Wave encourages early stopping. This contradiction provides an interesting comparison point in our additional benchmark. The results are shown in Table 1:
>
> | Method         | Acc          | Prec.           | Recall           | F1 Score          |
> |----------------|--------------|--------------|--------------|--------------|
> | CE             | 95.77 ± 0.35 | 83.31 ± 4.02 | **75.20 ± 2.78** | **76.51 ± 2.99** |
> | SAM [1]            | 94.49 ± 0.29 | 77.34 ± 1.69 | 69.70 ± 2.16 | 70.55 ± 1.89 |
> | PLM [2]           | **96.70 ± 0.06** | 79.22 ± 1.75| 72.81 ± 1.81 | 74.09 ± 1.65 |
> | SDN [3]           | 93.91 ± 1.13 | 75.92 ± 5.02 | 69.33 ± 2.55 | 69.80 ± 3.60 |
> | CE-label Wave [4] | 95.21 ± 0.23 | **84.23 ± 2.78** | 73.66 ± 2.15 | 75.29 ± 2.34 |
> Table 1. The accurancy, precision, recall and F1 Score of each methods with ResNet-50 backbone.
>
> Our expanded benchmark yields several interesting observations. Most significantly, Part-Level Multilabeling (PLM) achieved the highest accuracy of 96.70%. PLM's success suggests that partial features extracted by cropping the images are indeed helpful for correcting labels of noisy instances in our fine-grained classification task. In the early stop versus late stop debate, our results favor early stopping. CE-label Wave, which implements early stopping, maintains performance on par with standard CE, while SDN, advocating for late stopping, underperforms. This may imply that the memorization effect is still dominant in our dataset, and early intervention is beneficial for training robust models. Contrary to recent analyses [5], SAM did not provide superior results in our benchmark, underscoring the unique challenges presented by our dataset. It's worth noting that despite PLM's superior performance, the gap between it and standard CE remains relatively small (less than 1% in accuracy). This reinforces our original observation that the fine-grained, imbalanced nature of the Noisy Ostracods dataset presents significant challenges to current robust learning techniques.
>
>
>
> # References
>
> [1] Foret, P., Kleiner, A., Mobahi, H., & Neyshabur, B. (2021). Sharpness-aware Minimization for Efficiently Improving Generalization. In International Conference on Learning Representations.
> [2] Zhao, R., Shi, B., Ruan, J., Pan, T., & Dong, B. (2024). Estimating Noisy Class Posterior with Part-level Labels for Noisy Label Learning. In Conference on Computer Vision and Pattern Recognition.
> [3] Bai, Y., Han, Z., Yang, E., Yu, J., Han, B., Wang, D., & Liu, T. (2023). Subclass-Dominant Label Noise: A Counterexample for the Success of Early Stopping. In Thirty-seventh Conference on Neural Information Processing Systems.
> [4] Yuan, S., Feng, L., & Liu, T. (2024). Early Stopping Against Label Noise Without Validation Data. In The Twelfth International Conference on Learning Representations.
> [5] Baek, C., Kolter, J. Z., & Raghunathan, A. (2024). Why is SAM Robust to Label Noise? In The Twelfth International Conference on Learning Representations.

---

> > ### Comment · Reviewer_x9Zo · 2024-08-20
> > **Response to Rebuttal by Authors**
> >
> > The rebuttal addresses most of the weaknesses. The reviewer raises the rating from 5 to 7 and recommends that
> > - prepare a detailed tutorial/README for using this dataset and benchmark.
> > - proof-reading for the final manuscript.
> >
> > Good luck with your submission.

---

> ### Author Rebuttal · Authors · 2024-08-17
>
> ## (Reply to reviewer 2/3)
>
> **Implementation Details:**
>
> We implemented the additional methods as follows, with all modifications available in our code repository:
>
> SAM [1]: We followed Algorithm 1 from the original paper, setting p=1, q=1, and ρ=0.04 as per the recommended hyperparameters. The implementation is in train_engine.py of our code repository.
> PLM [2]: We adapted the official implementation, modifying the anchor point per class to 1 to accommodate classes with only one image. Other parameters follow the official implementation. The modified version is available in our repository under PLM_mod.
> SDN [3]: Instead of training the model using the official implementation, we provided the trained weights from CE and then  applied DBSCAN. The modified version is available in our repository under SDN_mod.
> Label Wave [4]: We implemented this method in train_engine.py, setting patience=5 and k=3.
> Due to the time constraints of the rebuttal period, we were unable to perform extensive validation on these implementations. Consequently, some results may be subject to change upon further verification and optimization.
>
> # 2. Implementating additional label cleanning methods.
>
> As per the reviewer's suggestion, we implemented two additional label cleaning methods mentioned in the AQuA [8] paper: Area Under Margin (AUM) [6] and Contrastive and Influential Counterexample Strategy (CINCER) [7]. The results are presented in Table 2.
> | Method | Hit Rate | Hit Rate Feature Error | Hit Rate Label Error | Fix Precision | Found Pseudo Class | P. Hit Rate | Hit Rate w/o P. |
> |--------|----------|------------------------|----------------------|---------------|-------------------|-------------|-----------------|
> | CL [9]     | 59.37%   | 62.95%                 | 54.06%               | 64.13%        | F                 | **7.94%**   | 75.41%          |
> | NECV  | **71.19%** | **80.42%**           | **57.50%**           | 55.87%        | T                 | 5.82%       | **91.58%**      |
> | AUM [6]   | 29.56%   | 24.84%                 | 36.56%               | **86.72%**    | F                 | 3.17%       | 37.79%          |
> | CINCER [7] | 53.71%   | 56.63%                 | 49.38%               | 54.60%        | T                 | 4.76%       | 68.98%          |
>
> Table 2. Label fixing comparsion.
>
> Our analysis of these additional methods yields several insights:
> * Hit Rate Performance: In terms of hit rate, which indicates how many label errors a model can identify, both AUM and CINCER underperformed compared to our baseline methods, Naive Ensembling Cross Validation (NECV) and Confident Learning (CL) [9]. NECV remains the top performer with a hit rate of 71.19%, significantly outpacing AUM (29.56%) and CINCER (53.71%).
> * Feature vs. Label Errors: NECV shows superior performance in detecting both feature errors (80.42%) and label errors (57.50%). This suggests that our ensemble approach is robust across different types of noise in our dataset.
> Precision Trade-off: AUM achieved the highest fix precision at 86.72%, substantially outperforming other methods in this metric. However, this high precision comes at the cost of a much lower hit rate, indicating a trade-off between precision and recall in label error detection.
> * Pseudo-class Detection: NECV and CINCER successfully identified the pseudo-class *Cyperdies*, while CL and AUM failed to do so. This highlights the importance of methods that can detect novel or erroneous classes in fine-grained classification tasks.
> * Performance without *Paradoxostomatid* (P.): When excluding the challenging Paradoxostomatid class, all methods show improved performance, with NECV achieving a remarkable 91.58% hit rate. This result is consistent with our paper's findings: none of the methods could effectively deal with majority-is-wrong errors for certain classes. The *Paradoxostomatid* class exemplifies this challenge, where the majority of labels are incorrect, severely impacting the performance of all label cleaning methods.
> Overall, while AUM offers high precision and CINCER provides balanced performance similar to CL, our baseline NECV method continues to demonstrate superior overall performance in detecting label errors in our complex, fine-grained dataset. These results is enhancing one of the original goal of proposal of the noisy ostracods dataset: calling for more attention on label correction method on Label Noise Learning research.
>
> **Implementation details** AUM was implemented by injecting negative classes into the majority classes and calculating the margin as the original paper suggests. Detailed implementation is inside `train_engine.py`. For CINCER we are just interested in finding suspicious labels, we did not implement the Fisher Information matrix part to suggest possible correct label. Detailed implementation is within `evaluation.py`.
>
> # References
> [6] Geoff Pleiss, Tianyi Zhang, Ethan Elenberg, and Kilian Q. Weinberger. Identifying mislabeled
> data using the area under the margin ranking. In Proceedings of the 34th International Conference
> on Neural Information Processing Systems.
> [7] Teso, S., Bontempelli, A., Giunchiglia, F., & Passerini, A. (2021). Interactive Label Cleaning with Example-based Explanations. In A. Beygelzimer, Y. Dauphin, P. Liang, & J. Wortman Vaughan (Eds.), Advances in Neural Information Processing Systems.
> [8] Mononito Goswami et al. “AQuA: A Benchmarking Tool for Label Quality Assessment”. In: Advances in Neural Information Processing Systems.
> [9] Curtis G. Northcutt, Lu Jiang, and Isaac L. Chuang. “Confident Learning: Estimating Uncer351
> tainty in Dataset Labels”. In: Journal of Artificial Intelligence Research (JAIR)

---

> ### Author Rebuttal · Authors · 2024-08-17
>
> ## (Reply to reviewer 3/3)
> # 3. Addressing Reproducibility and Maintainability Issues
> ## 3.1 Model Weights
> We acknowledge the importance of releasing model weights for reproducibility. We will share links to approximately 300 model weights via an official comment on this OpenReview page after addressing all reviewer feedback. Due to challenges in our project management, we need to run inference on these models to identify which produced specific results. This process will take some time, and we'll gradually populate the shared folder with verified models.
>
> ## 3.2 GitHub Code Repository
> We have made our GitHub repository public. After careful analysis, we've identified several areas for improvement:
>
> 1. Code Integration: Our attempt to minimize changes to original implementations resulted in a patchwork of code from different authors. We plan to refactor the codebase into a standardized trainer -> train engine structure, with parameters controlled via config.yaml.
> 2. Method Incompatibilities: Label correction and robust training methods have different input requirements. We're working on a unified interface to handle these differences more elegantly.
> 3. Documentation: We've identified missing crucial components like requirements.txt. We're actively addressing these oversights.
>
> Given these challenges, we've decided to upload our modifications of SDN[3] and PLM[2] to separate repositories. Our immediate focus has been on implementing and testing state-of-the-art Learning with Label Noise methods, which has delayed comprehensive code refactoring. We're committed to gradual improvements, taking inspiration from the Uncertainty Baselines [10] official implementation for best practices.
>
> ## 3.3 Leaderboard
> We've included the results table from our paper as an initial leaderboard. We're also exploring the creation of a Papers with Code page for our dataset to facilitate community engagement and tracking of progress.
>
> ## 3.4. Proofreading and Typo Correction
> We've conducted two rounds of revisions with all co-authors, significantly reducing typos and improving overall clarity.
>
> We sincerely appreciate the reviewer's thorough evaluation and insightful comments. Hope the above discussion adressed the reviewer's concerns. We remain committed to improving our work based on the valuable feedback received and look forward to continuing this important line of research.
>
>
> # References
> [10] Z. Nado, N. Band, M. Collier, J. Djolonga, M. Dusenberry, S. Farquhar, A. Filos, M. Havasi, R. Jenatton, G. Jerfel, J. Liu, Z. Mariet, J. Nixon, S. Padhy, J. Ren, T. Rudner, Y. Wen, F. Wenzel, K. Murphy, D. Sculley, B. Lakshminarayanan, J. Snoek, Y. Gal, and D. Tran. Uncertainty Baselines: Benchmarks for uncertainty & robustness in deep learning, arXiv preprint arXiv:2106.04015, 2021.

---

> ### Author Response · Authors · 2024-08-20
> **Response to Reiviewer**
>
> We are delighted to hear the concern was mostly adressed!
> We will keep working based on your advices!

---

### Author Response · Authors · 2024-08-17
**Links to Data, model and additional implementations**

Models (Gradually updating):

https://connecthkuhk-my.sharepoint.com/:f:/g/personal/jiamianh_connect_hku_hk/EttBGfh88W1EuvyTTDDiUSsBvgZbDLmDcQrkqrXnW-Qmbw?e=iPPx9k

Data:

https://connecthkuhk-my.sharepoint.com/:u:/g/personal/jiamianh_connect_hku_hk/ETA33LiJZyhGukjzHDM69mYBv25Y2XvlEe8EVgAu2_T4HQ?e=f5DJ06

2022 Data:

https://connecthkuhk-my.sharepoint.com/:u:/g/personal/jiamianh_connect_hku_hk/EYP9K2PASn5OhK2vBJA628UBwsRvBrKTNJFai5uvsqxCmw?e=suhXwo

All labels: (updated)

https://connecthkuhk-my.sharepoint.com/:u:/g/personal/jiamianh_connect_hku_hk/EROmG_7pjBxJsxOIQfmCqbUB46f6gw6_etGaWzeP21hvHw?e=KIrhsI

PLM_modified:

https://github.com/H-Jamieu/PLM_mod

SDN_modified:

https://github.com/H-Jamieu/SDN_mod


If any link not working, just comment to me.

---

### Author Response · Authors · 2024-09-01
**Summary of replies to reviewers**

Thank you to all reviewers for their careful and thoughtful feedback. We have addressed the main concerns as follows:

1. GitHub Repository: The link is now public and accessible.
2. State-of-the-Art (SOTA) Methods: We have implemented four additional Label Noise Learning methods (mostly from 2023-2024) and two more label fix methods. Further implementations are in progress.
3. Dataset Accessibility: We have updated the documentation with publicly accessible links to the dataset.
4. Code Usability: We are currently refactoring the code to improve user-friendliness. This process will be completed after implementing one additional LNL method.
5. Model Availability: Download links for the models are now provided.
6. Proofreading: Two rounds of revision have been completed, with two more currently underway.

**Expanded Results:**

1. **State-of-the-Art (SOTA) Methods**: We've implemented four additional Label Noise Learning methods (2023-2024) and two more label fix methods. Table 1 has been expanded to include these new methods:

| Backbone | Resnet-50 |  |  |  | Vit-b-16 |  |  |  |
|----------|-----------|-----------|--------|----------|----------|-----------|--------|----------|
| Methods  | Accuracy  | Precision | Recall | F1-score | Accuracy | Precision | Recall | F1-score |
| CE       | 95.98%    | 88.50%    | 77.80% | 79.51%   | 95.03%   | 83.31%    | 75.30% | 76.64%   |
| CE+mixup | 95.11%    | 74.96%    | 69.83% | 69.42%   | 90.33%   | 57.99%    | 51.95% | 52.98%   |
| CL       | 95.16%    | 80.97%    | 69.01% | 71.51%   | 90.62%   | 57.48%    | 53.27% | 53.75%   |
| Loss-clip| 95.79%    | 81.69%    | 72.78% | 74.09%   | 93.71%   | 72.14%    | 64.27% | 65.26%   |
| Co-teaching | 95.79% | 79.01%    | 71.79% | 73.23%   | 94.71%   | 77.77%    | 71.69% | 72.37%   |
| Co-teaching+ | 96.19%| 84.09%    | 77.57% | 78.21%   | 91.82%   | 66.04%    | 63.45% | 63.65%   |
| Divide-mix | 95.50%  | 53.42%    | 57.33% | 54.86%   | 84.79%   | 23.00%    | 28.22% | 24.92%   |
| SAM      | 94.84%    | 78.93%    | 68.08% | 70.60%   | 93.21%   | 77.50%    | 64.78% | 67.75%   |
| PLM      | 96.77%    | 77.77%    | 72.74% | 73.46%   | 95.26%   | 75.32%    | 69.66% | 71.10%   |
| SDN      | 95.11%    | 78.13%    | 71.31% | 72.04%   | 92.84%   | 76.49%    | 62.16% | 64.96%   |
| LW       | 95.50%    | 86.59%    | 74.28% | 76.93%   | 94.65%   | 77.15%    | 69.61% | 71.35%   |


2. **Label Correction**: Following reviewer x9Zo's advice, we've expanded Table 2:

| Method | Hit rate | Hit rate feature error | Hit rate label error | Fix precision | Found pseudo class | P.Hit rate | Hit rate w/o P. |
|--------|----------|------------------------|----------------------|---------------|---------------------|------------|-----------------|
| CL | 59.37% | 62.95% | 54.06% | 64.13% | F | 7.94% | 75.41% |
| Simifeat-CLIP | 26.29% | 30.11% | 20.63% | 17.13% | F | 6.88% | 32.34% |
| Simifeat-CLIP-mv | 26.29% | 29.05% | 22.19% | 17.29% | F | 7.41% | 32.18% |
| Simifeat-MAE-trained | 28.18% | 30.32% | 25.00% | 47.16% | T | 3.17% | 35.97% |
| Simifeat-MAE-trained-mv | 31.07% | 32.21% | 29.38% | 44.67% | T | 2.65% | 39.93% |
| Simifeat-MAE-raw | 27.55% | 32.21% | 20.63% | 13.83% | T | 12.70% | 32.18% |
| Simifeat-MAE-raw-mv | 26.79% | 32.00% | 19.06% | 13.87% | T | 14.29% | 30.69% |
| NECV | 71.19% | 80.42% | 57.50% | 55.87% | T | 5.82% | 91.58% |
| AUM | 29.56% | 24.84% | 36.56% | 86.72% | F | 3.17% | 37.79% |
| CINCER | 53.71% | 56.63% | 49.38% | 54.60% | T | 4.76% | 68.98% |

**Error Correction:**

We identified an error in the `ostracods_genus_clean_test.csv` file shared online. The file mistakenly eliminated `rawSample_B1b_51_ind1.tif`, potentially causing a ~0.02% difference in all metrics during reproducibility checks. We have fixed this error and will update the link to the corrected label file.
Thanks all reviewers for their constructive and thought feedback, our experiments do looks stronger now.

**Ongoing Work:**

Further experiments are in progress, and we look forward to sharing additional results by the due date.

We appreciate the constructive feedback, which has significantly strengthened our experiments and overall contribution.

---

### Decision · Program_Chairs · 2024-09-26

**Decision:**

Accept (Poster)

**Comment:**

This submission received four ratings (7, 7, 6 and 7), averaging 6.75, which is a positive score. After rebuttal, all reviewers have shown that their concerns have been mostly addressed and two reviewers gave some minor suggestions for further improvement. Overall, the submission is totally well supported given the dataset motivation and experimental analysis. After carefully checking the concerns of all reviewers and the authors' rebuttal, I suggest the acceptance. Hope the authors incorporate the suggestions (including further suggestions) into the final revision.